# The transcriptional corepressor CTBP-1 acts with the SOX family transcription factor EGL-13 to maintain AIA interneuron cell identity in *Caenorhabditis elegans*

Josh Saul, Takashi Hirose†, H Robert Horvitz*

Department of Biology, Massachusetts Institute of Technology, Howard Hughes Medical Institute, Cambridge, United States

*For correspondence: horvitz@mit.edu

Present address: †Sysmex Corporation, 4-4-4 Takatsukadai, Nishi-ku, Kobe, Japan

**Competing interest:** The authors declare that no competing interests exist.

**Abstract** Cell identity is characterized by a distinct combination of gene expression, cell morphology, and cellular function established as progenitor cells divide and differentiate. Following establishment, cell identities can be unstable and require active and continuous maintenance throughout the remaining life of a cell. Mechanisms underlying the maintenance of cell identities are incompletely understood. Here, we show that the gene *ctbp-1,* which encodes the transcriptional corepressor *C-*terminal *b*inding *p*rotein-1 (CTBP-1), is essential for the maintenance of the identities of the two AIA interneurons in the nematode *Caenorhabditis elegans. ctbp-1* is not required for the establishment of the AIA cell fate but rather functions cell-autonomously and can act in later larval stage and adult worms to maintain proper AIA gene expression, morphology and function. From a screen for suppressors of the *ctbp-1* mutant phenotype, we identified the gene *egl-13,* which encodes a SOX family transcription factor. We found that *egl-13* regulates AIA function and aspects of AIA gene expression, but not AIA morphology. We conclude that the CTBP-1 protein maintains AIA cell identity in part by utilizing EGL-13 to repress transcriptional activity in the AIAs. More generally, we propose that transcriptional corepressors like CTBP-1 might be critical factors in the maintenance of cell identities, harnessing the DNA-binding specificity of transcription factors like EGL-13 to selectively regulate gene expression in a cell-specific manner.

## Editor's evaluation

The paper presents an interesting addition to our understanding of cell fate maintenance, making incisive use of the power of *C. elegans* genetics.

## Introduction

Over the course of animal development, complex networks of transcription factors act and interact to drive the division and differentiation of progenitor cells toward terminal cell identities (***Davidson et al., 1998***; ***Davidson et al., 2002***; ***Levine and Davidson, 2005***; ***Hobert, 2016a***; ***Hsieh and Zhao, 2016***; ***Homem et al., 2015***; ***Altun-Gultekin et al., 2001***; ***Zeng and Sanes, 2017***). These networks of transcriptional activity often culminate in the activation of master transcriptional regulators that are responsible for directing the differentiation of a diverse range of cell and tissue types (***Hobert, 2016a***; ***Baker, 2001***; ***Deneris and Hobert, 2014***; ***Hobert et al., 2010***; ***Masoudi et al., 2018***). Examples of such master transcriptional regulators include the mammalian bHLH transcription factor MyoD, which

specifies skeletal muscle cells (*Weintraub et al., 1991*; *Lassar, 2017*; *Wardle, 2019*); the *Drosophila* Pax-family transcription factor Eyeless, which drives differentiation of the fly eye (*Halder et al., 1995*; *Gehring, 1996*; *Shen and Mardon, 1997*; *Treisman, 2013*; *Lima Cunha et al., 2019*); and the *C. elegans* GATA transcription factor ELT-2, essential for development of the worm intestine (*Fukushige et al., 1998*; *Fukushige et al., 1999*; *McGhee et al., 2009*; *Block and Shapira, 2015*). Many such master transcriptional regulators are not only required to establish the identities of specific cell types but are subsequently continuously required to maintain those identities for the remaining life of the cell (*Hobert, 2016a*; *McGhee et al., 2009*; *Matson et al., 2011*; *Mall et al., 2017*; *Simon et al., 2004*; *Vissers et al., 2018*; *Hsiao et al., 2013*). Defects in the maintenance of cell identities can manifest as late-onset misregulated gene expression, altered morphology or disrupted cellular function, and often become progressively worse as the cell ages (*Matson et al., 2011*; *Vissers et al., 2018*; *Riddle et al., 2013*; *O'Meara et al., 2010*; *Xu et al., 2017*).

Previous studies of the nematode *Caenorhabditis elegans* have identified a class of master transcriptional regulators, termed terminal selectors (*Hobert, 2016a*; *Masoudi et al., 2018*; *Hobert, 2008*; *Hobert, 2011*; *Hobert and Kratsios, 2019*; *Hobert, 2016b*; *Zhang et al., 2014*). Terminal selectors drive the expression of whole batteries of gene activity that ultimately define the unique features of many different cell types (*Deneris and Hobert, 2014*; *Hobert et al., 2010*; *Masoudi et al., 2018*). Individual terminal selectors have been shown to contribute to the establishment and maintenance of multiple distinct *C. elegans* cell types and to drive the expression of many cell-type-specific genes (*Altun-Gultekin et al., 2001*; *Zhang et al., 2014*; *Serrano-Saiz et al., 2013*; *Duggan et al., 1998*; *Kim et al., 2015*; *Alqadah et al., 2015*). However, it has been unclear how individual terminal selectors can drive the expression of cell-type-specific genes in only the appropriate cell types rather than in all cells in which they act (*Kerk et al., 2017*; *Zhou and Walthall, 1998*; *Winnier et al., 1999*). Recent work has shown that terminal selectors appear to broadly activate the expression of many genes, including cell-type-specific genes, in all cells in which they function (*Kerk et al., 2017*; *Yu et al., 2017*). Piecemeal assemblies of transcription factors are then responsible for pruning this broad expression to restrict expression of cell-type-specific genes to the appropriate cell types (*Kerk et al., 2017*; *Yu et al., 2017*). This restriction of the activation of gene expression by terminal selectors appears to be an essential aspect of proper cell-identity maintenance (*Vissers et al., 2018*; *Kerk et al., 2017*; *Yu et al., 2017*; *Wyler et al., 2016*). However, it is not known how the myriad of transcription factors utilized to restrict terminal selector gene activation are coordinated and controlled.

Here we report the discovery that the *C. elegans* gene *ctbp-1,* which encodes the sole worm ortholog of the *C*-terminal *B*inding *P*rotein (CtBP) family of transcriptional corepressors (*Turner and Crossley, 2001*; *Chinnadurai, 2002*; *Chinnadurai, 2003*; *Shi et al., 2003*; *Stankiewicz et al., 2014*; *Nicholas et al., 2008*; *Reid et al., 2014*; *Reid et al., 2015*; *Sherry et al., 2020*), functions to maintain the cell identity of the two AIA interneurons. We demonstrate that CTBP-1 functions with the SOX-family transcription factor EGL-13 (*Gramstrup Petersen et al., 2013*; *Cinar et al., 2003*) to maintain multiple aspects of the AIA cell identity and propose that CTBP-1 does so in part by utilizing EGL-13 to repress transcriptional activity in the AIAs.

## Results

### Mutations in *ctbp-1* cause *ceh-28* reporter misexpression in the AIA neurons

In previous studies, we screened for and characterized mutations that prevent the programmed cell death of the sister cell of the *C. elegans* M4 neuron (*Hirose and Horvitz, 2013*; *Hirose et al., 2010*). For these screens, we used the normally M4-specific GFP transcriptional reporter $P_{ceh-28}::gfp$ and identified isolates with an undead M4 sister cell, which expresses characteristics normally expressed by the M4 cell, on the basis of ectopic GFP expression. In addition to mutants with an undead M4 sister cell, we isolated 18 mutant strains that express $P_{ceh-28}::gfp$ in a manner uncharacteristic of M4 or its undead sister. These mutants express $P_{ceh-28}::gfp$ in a bilaterally symmetric pair of cells located near the posterior of the *C. elegans* head, far from both M4 and the single M4 sister cell (*Figure 1A*).

These mutations define a single complementation group, and all 18 mutant strains have mutations in the transcriptional corepressor gene *ctbp-1* (*Figure 1C*; *Figure 1—figure supplement 1A-B*; *Figure 1—figure supplement 2A*). These *ctbp-1* alleles include three splice-site mutations and nine

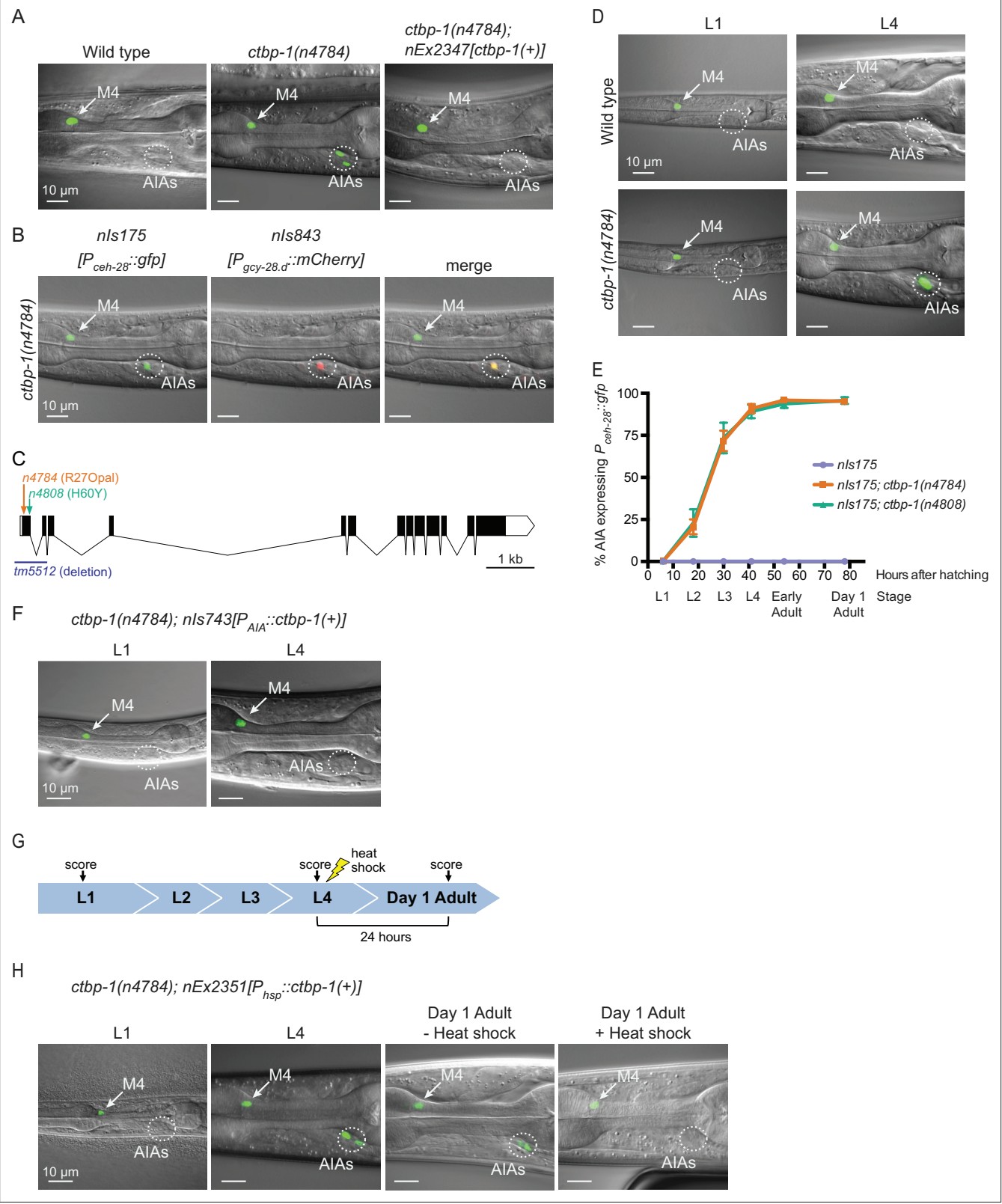

**Figure 1.** *ctbp-1* mutants misexpress $P_{ceh-28}::gfp$ in the AIA neurons. (**A**) Expression of the M4-specific marker *nIs175[$P_{ceh-28}$::gfp]* in the wild type (left panel), a *ctbp-1(n4784)* mutant (middle panel), and a *ctbp-1* mutant carrying an extrachromosomal array expressing wild-type *ctbp-1* under its native promoter (*nEx2347*) (right panel). Arrow, M4 neuron. Circle, AIAs. Scale bar, 10 μm. (**B**) A *ctbp-1(n4784)* mutant expressing *nIs175* (left panel) and the AIA marker *nIs843[$P_{gcy-28.d}$::mCherry]* (middle panel). Merge, right panel. Arrow, M4 neuron. Circle, AIAs. Scale bar, 10 μm. (**C**) Gene diagram of the *ctbp-1a*

*Figure 1 continued on next page*

*Figure 1 continued*

isoform. Arrows (above), point mutations. Line (below), deletion. Scale bar (bottom right), 1 kb. Additional *ctbp-1* alleles are shown in **Figure 1—figure supplement 1B**. (**D**) *nIs175* expression in wild-type (top) and *ctbp-1(n4784)* (bottom) worms at the L1 larval stage (left) and L4 larval stage (right). Arrow, M4 neuron. Circle, AIAs. Scale bar, 10 µm. (**E**) Percentage of wild-type, *ctbp-1(n4784)*, and *ctbp-1(n4808)* worms expressing *nIs175* in the AIA neurons over time. Time points correspond to the L1, L2, L3, and L4 larval stages, early adult, and day 1 adult worms (indicated below X axis). Mean ± SEM. n ≥ 60 worms scored per strain per stage, four biological replicates. (**F**) Expression of *nIs175* in *ctbp-1* mutants containing a transgene driving expression of wild-type *ctbp-1* under an AIA-specific promoter (*nIs743[P_{gcy-28.d}::ctbp-1(+)]*) in L1 and L4 larval worms. Arrow, M4 neuron. Circle, AIAs. Scale bar, 10 µm. (**G**) Schematic for the heat-shock experiment shown in **H**. (**H**) *nIs175* expression in *ctbp-1(n4784)* mutants carrying the heat-shock-inducible transgene *nEx2351[P_{hsp-16.2}::ctbp-1(+); P_{hsp-16.41}::ctbp-1(+)]*. Arrow, M4 neuron. Circle, AIAs. Scale bar, 10 µm. All strains shown contain the transgene *nIs175[P_{ceh-28}::gfp]*. Images are oriented such that left corresponds to anterior, top to dorsal. Quantification of reporter expression from **A, B, F** in **Figure 1—figure supplement 2**. Quantification of reporter expression from **H** in **Figure 1—figure supplement 2**.

The online version of this article includes the following source data and figure supplement(s) for figure 1:

**Source data 1.** Source data for **Figure 1** and supplements.

**Figure supplement 1.** Additional *ctbp-1* mutant alleles cause misexpression of *P_{ceh-28}::gfp* in the AIA neurons.

**Figure supplement 2.** Quantification of *ctbp-1* strains misexpressing *P_{ceh-28}::gfp*.

---

nonsense mutations (such as the mutation *n4784*, an early nonsense mutation and one of many presumptive null alleles of the gene). The mutant phenotype is recessive, and a transgenic construct carrying a wild-type copy of *ctbp-1* expressed under its native promoter fully rescued the GFP misexpression caused by *n4784* (**Figure 1A**; quantified in **Figure 1—figure supplement 2A**). *tm5512*, a 632 bp deletion spanning the transcription start site and first two exons of the *ctbp-1a* isoform and a presumptive null allele of this gene (*C. elegans Deletion Mutant Consortium, 2012*), likewise caused *P_{ceh-28}::gfp* misexpression in two cells in the posterior region of the head (**Figure 1—figure supplement 1C-D**), similar to our *ctbp-1* isolates. These findings demonstrate that loss of *ctbp-1* function is responsible for *P_{ceh-28}::gfp* misexpression.

To determine the identity of the cells misexpressing the normally M4-specific marker *P_{ceh-28}::gfp*, we examined reporters for cells in the vicinity of the observed misexpression in *ctbp-1* mutants. The AIA-neuron reporter *nIs843[P_{gcy-28.d}::mCherry]* showed complete overlap with misexpressed *P_{ceh-28}::gfp*, indicating that the cells misexpressing the M4 reporter are the two bilaterally symmetric and embryonically-generated AIA interneurons (**Figure 1B**).

## The penetrance of *ceh-28* reporter misexpression in the AIA neurons increases with age

While characterizing *ctbp-1* mutants, we noticed that fewer young worms misexpress *P_{ceh-28}::gfp* in the AIAs than do older worms (**Figure 1D**). To investigate the temporal aspect of this phenotype, we scored *ctbp-1* mutants for *P_{ceh-28}::gfp* misexpression throughout the four worm larval stages (L1-L4) and into the first day of adulthood ('early' and 'day 1' adults). *ctbp-1* mutants rarely misexpressed *P_{ceh-28}::gfp* at early larval stages, but displayed an increasing penetrance, though invariant expressivity, of this defect as worms transitioned through larval development, such that by the last larval stage (L4) nearly all worms exhibited reporter misexpression specifically and solely in the AIAs (**Figure 1E**). A similar stage-dependent increase in reporter expression in *ctbp-1* mutants occurred in mutants carrying a second independently generated *ceh-28* reporter, *nIs348[P_{ceh-28}::mCherry]* (**Figure 1—figure supplement 1E**). These results demonstrate that *ctbp-1* function prevents an age-dependent misexpression of the M4-specific gene *ceh-28* in the unrelated AIA neurons.

We next asked in what cells and at what stages *ctbp-1* functions to suppress *P_{ceh-28}::gfp* expression in the AIAs. We generated a transgenic construct that expresses wild-type *ctbp-1* specifically in the AIAs, *nIs743[P_{gcy-28.d}::ctbp-1(+)]* (hereafter referred to as *nIs743[P_{AIA}::ctbp-1(+)]*). We found that AIA-specific restoration of *ctbp-1* was sufficient to suppress *P_{ceh-28}::gfp* misexpression in an otherwise *ctbp-1* mutant background (**Figure 1F**; quantified in **Figure 1—figure supplement 2A**), demonstrating that *ctbp-1* is able to act cell-autonomously to regulate *ceh-28* expression in the AIA neurons.

To determine if *ctbp-1* can act in older animals to suppress AIA gene misexpression, we generated a transgenic construct that drives expression of wild-type *ctbp-1* throughout the worm in response to a short heat shock, *nEx2351[P_{hsp-16.2}::ctbp-1(+); P_{hsp-16.41}::ctbp-1(+)]* (hereafter referred to as *nEx2351[P_{hsp}::ctbp-1(+)]*). We found that heat shock during the L4 larval stage was sufficient to suppress *P_{ceh-28}::gfp* misexpression in adult *ctbp-1* mutant AIAs (**Figure 1G–H**; quantified in **Figure 1—figure**

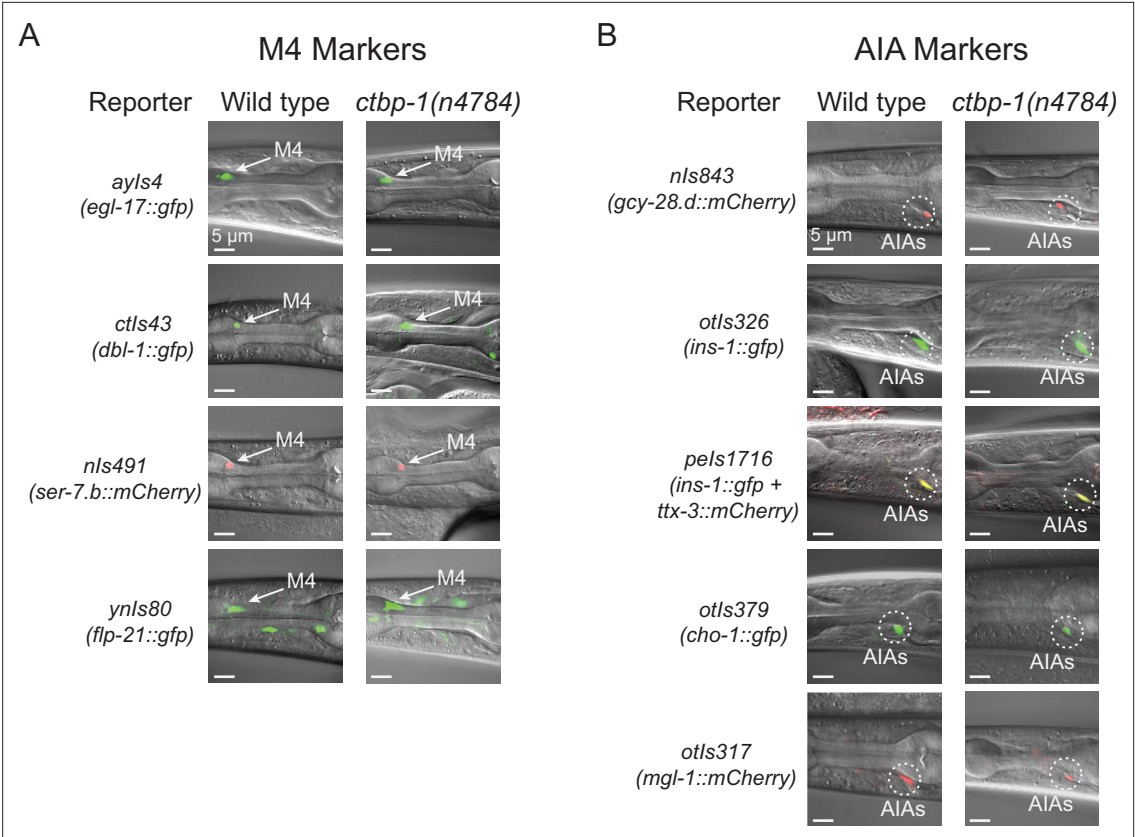

**Figure 2.** *ctbp-1* mutant AIAs retain multiple aspects of their AIA gene expression profile. (**A–B**) Expression of (**A**) M4 markers *egl-17*, *dbl-1*, *ser-7.b*, and *flp-21* and (**B**) AIA markers *gcy-28.d*, *ins-1*, *ttx-3*, *cho-1*, and *mgl-1* in wild-type (left image) and *ctbp-1(n4784)* (right image) L4 larval worms. Arrow, M4 neuron. Circles, AIAs. Scale bar, 5 μm. Images are oriented such that left corresponds to anterior, top to dorsal. Quantification of reporter expression in *Figure 2—figure supplement 1A-B*.

The online version of this article includes the following source data and figure supplement(s) for figure 2:

**Source data 1.** Source data for *Figure 2* and supplements.

**Figure supplement 1.** Quantification of M4 and AIA marker expression.

*supplement 2B*), demonstrating that *ctbp-1* can act in L4-to-young adult stage worms to regulate AIA gene expression.

From these data we conclude that *ctbp-1* is able to act cell-autonomously and in L4-to-young adult worms to prevent expression of at least one non-AIA gene in the AIA neurons.

## *ctbp-1* mutant AIAs are not transdifferentiating into an M4-like cell identity

We asked if $P_{ceh-28}::gfp$ misexpression in the AIAs of *ctbp-1* mutants might be a consequence of the AIAs transdifferentiating into an M4-like cell identity. We scored *ctbp-1* mutants for cell-type markers expressed in, although not necessarily unique to, either M4 or the AIA neurons (*Figure 2A–B*; quantified in *Figure 2—figure supplement 1A-B*). We found that *ctbp-1* mutant AIAs expressed all five of five AIA markers tested and did not express any of four other (non-*ceh-28*) M4 markers tested. Of particular note, *ctbp-1* mutant AIAs did not misexpress either of the two tested M4 genes known to be directly regulated by *ceh-28* (i.e. *dbl-1* and *egl-17*), indicating that the *ceh-28* misexpression in mutant AIAs does not activate the *ceh-28* regulatory pathway (*Ramakrishnan and Okkema, 2014*; *Ramakrishnan et al., 2014*). We conclude that *ctbp-1* mutant AIAs are not transdifferentiated into M4-like cells and instead seem to retain much of their AIA identity while gaining at least one M4 characteristic (i.e. *ceh-28* expression) later in life.

## *ctbp-1* mutants display an increasingly severe disruption of AIA morphology

Because of the time-dependency of the defect of *ctbp-1* mutants in AIA cell identity, we hypothesized that *ctbp-1* might act to maintain the AIA cell identity. To test this hypothesis, we examined morphological and functional aspects of AIA identity at both early (L1) and late (L4) larval stages. To assay AIA morphology, we generated a transgenic construct driving expression of GFP throughout the AIA cell (*nIs840[P_{gcy-28.d}::gfp]*). We crossed this construct into *ctbp-1* mutant worms and visualized AIA morphology in L1 and L4 larvae as well as in day 1 adults (*Figure 3A*). We found that L1 *ctbp-1* mutant AIAs appeared grossly wild-type in morphology (*Figure 3A*). However, L4 and adult *ctbp-1* mutant AIAs had ectopic neurite branches that extended from both the anterior and posterior ends of the AIA cell body (*Figure 3A*). The penetrance of these ectopic branches increased progressively in later larval stage and adult mutants (*Figure 3B–C*). Older *ctbp-1* mutant AIAs also appeared to have an elongated cell body compared to wild-type AIAs. Quantification of this defect revealed that L4 and adult mutant AIA cell bodies, but not those of L1s, were significantly longer than their wild-type counterparts (*Figure 3D*). To assess if this increase in AIA length was a consequence of an increase in AIA size, we measured the maximum area of the AIA cell body from cross-sections of these cells. We found that the maximum area of the AIA cell body did not significantly differ between wild-type and mutant AIAs at any stage (*Figure 3—figure supplement 1A*), indicating that mutant AIAs were misshapen but not enlarged. To confirm that we were not biased by an awareness of genotype while measuring AIA lengths, we blinded the wild-type and *ctbp-1* AIA images used for length measurements and scored the blinded images as either 'normal' or 'elongated' (*Figure 3—figure supplement 1B*). Again, at the L1 larval stage, both wild-type and *ctbp-1* mutant AIAs appeared overwhelmingly 'normal', whereas at both the L4 larval stage and in day 1 adults *ctbp-1* mutant AIAs were scored as 'elongated' at a consistently higher rate than their wild-type counterparts. Collectively, these results demonstrate that *ctbp-1* mutant AIAs display abnormal morphology and that the severity of the observed morphological defects in *ctbp-1* mutants increases from L1 to L4 to adulthood. Furthermore, the relative lack of AIA morphological defects in L1 *ctbp-1* mutants suggests that *ctbp-1* is not required for the establishment of proper AIA morphology but instead acts to maintain AIA morphology over time.

We next asked if *ctbp-1* acts cell-autonomously and at later stages to regulate AIA morphology as it does for AIA gene expression. We visualized *ctbp-1* mutant AIAs carrying the AIA-specific *ctbp-1(+)* rescue construct *nIs743[P_{AIA}::ctbp-1(+)]* (*Figure 3E*). We found that AIA-specific restoration of *ctbp-1* in mutant worms rescued all AIA morphological defects to near-wild-type levels at all stages tested, indicating that *ctbp-1* can act cell-autonomously to regulate AIA morphology (*Figure 3F–H*). Next, we visualized *ctbp-1* mutant worms carrying the heat shock-inducible *ctbp-1(+)* rescue construct *nEx2351[P_{hsp}::ctbp-1(+)]*. We found that heat shock at the L4 stage did not restore *ctbp-1* mutant AIA morphology in day 1 adults back to wild type. While heat-shocked adults did display a lower frequency of morphological defects than did their non-heat-shocked counterparts, these differences were not significant, suggesting that brief restoration of wild-type *ctbp-1* activity is not able to restore mutant AIA morphology (*Figure 3I–L*). To ensure that the heat shock itself had not caused the slight difference in the frequency of AIA morphological defects, we compared heat-shocked and non-heat-shocked wild-type and *ctbp-1* mutant worms for AIA morphology. We found no significant difference between heat-shocked and non-heat-shocked worms in the frequency of ectopic AIA projections or AIA length, indicating that brief heat shock does not appear to affect AIA morphology (*Figure 3—figure supplement 2A-C*). We speculate that the lack of restoration of morphology in late-stage worms might be a consequence of the defects being irreversible, and that *ctbp-1* might be continuously required to prevent such defects from occurring. From these data, we conclude that *ctbp-1* can act cell-autonomously, and possibly continuously, to maintain aspects of AIA morphology in a manner similar to AIA gene expression.

## *ctbp-1* mutants display a progressive decline of AIA function

The AIA interneurons integrate sensory information from a number of sensory neurons, resulting in modulation of the movement of the worm in response to environmental stimuli (*Tomioka et al., 2006*; *Iino and Yoshida, 2009*; *Shinkai et al., 2011*). The AIAs function in response to volatile odors and play an important role in learning associated with the sensation of volatile odors or salts (*Tomioka et al., 2006*; *Cho et al., 2016*). We asked if *ctbp-1* mutants are abnormal in a behavior known to require the

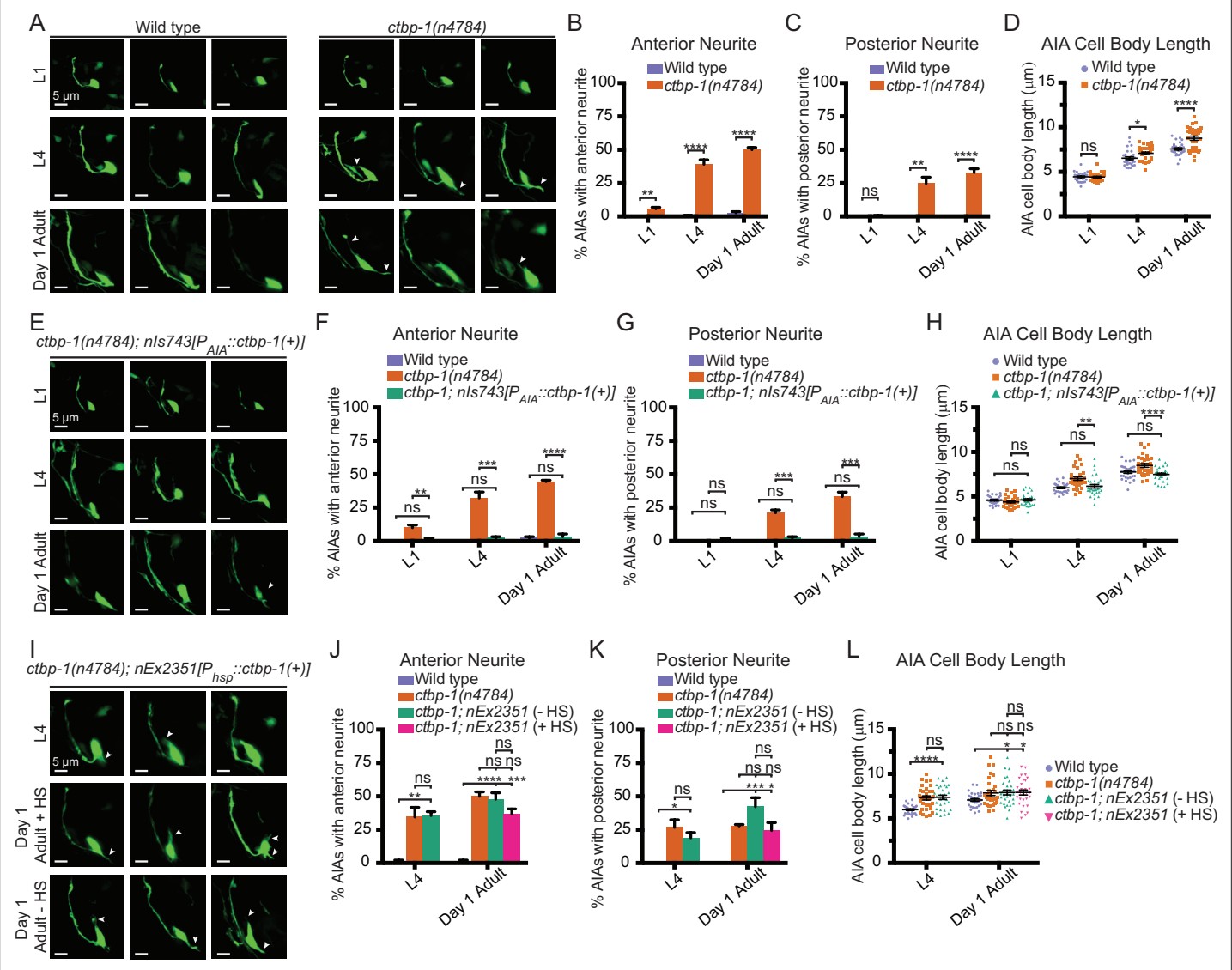

**Figure 3.** Loss of *ctbp-1* results in a progressive decline in AIA morphology. (**A**) Three representative images of an AIA neuron in wild-type (left) and *ctbp-1(n4784)* (right) worms at L1 (top), L4 (middle), and day 1 adult (bottom) stages. Arrows, examples of ectopic neurites protruding from the AIA cell body. Scale bar, 5 µm. (**B–C**) Percentage of AIAs in wild-type and *ctbp-1* worms at the L1, L4, and day 1 adult stages with an ectopic neurite protruding from the (**B**) anterior or (**C**) posterior of the AIA cell body. Mean ± SEM. n = 60 AIAs scored per strain per stage, four biological replicates. ns, not significant (p = 0.356), **p < 0.01, ****p < 0.0001, unpaired t-test. (**D**) Quantification of AIA cell body length in wild-type and *ctbp-1* worms at the L1, L4, and day 1 adult stages. Mean ± SEM. n = 30 AIAs scored per strain per stage. ns, not significant (p = 0.806), *p = 0.0133, ****p < 0.0001, unpaired t-test. (**E**) Three representative images of an AIA neuron in *ctbp-1; nIs743[P_{gcy-28.d}::ctbp-1(+)]* worms at L1 (top), L4 (middle), and day 1 adult (bottom) stages. Arrows, examples of ectopic neurites protruding from the AIA cell body. Scale bar, 5 µm. (**F–G**) Percentage of AIAs in wild-type, *ctbp-1,* and *ctbp-1; nIs743* worms at the L1, L4, and day 1 adult stages with an ectopic neurite protruding from the (**F**) anterior or (**G**) posterior of the AIA cell body. Mean ± SEM. n = 30 AIAs scored per strain per stage, three biological replicates. ns, not significant, **p = 0.0065, ***p < 0.001, ****p < 0.0001, one-way ANOVA with Tukey's correction. (**H**) Quantification of AIA cell body length in wild-type, *ctbp-1* and *ctbp-1; nIs743* worms at the L1, L4, and day 1 adult stages. Mean ± SEM. n ≥ 30 AIAs scored per strain per stage. ns, not significant, **p = 0.0015, ****p < 0.0001, one-way ANOVA with Tukey's correction. (**I**) Three representative images of an AIA neuron in *ctbp-1; nEx2351[P_{hsp-16.2}::ctbp-1(+); P_{hsp-16.41}::ctbp-1(+)]* worms at L4 (top), day 1 adult with heat shock (+ HS) (middle) and day 1 adult without heat shock (- HS) (bottom). Arrows, examples of ectopic neurites protruding from the AIA cell body. Scale bar, 5 µm. (**J–K**) Percentage of AIAs in wild-type, *ctbp-1* and *ctbp-1; nEx2351* worms at L4 and day 1 adult (with or without heat shock) stages with an ectopic neurite protruding from the (**J**) anterior or (**K**) posterior of the AIA cell body. Mean ± SEM. n = 30 AIAs scored per strain per stage, three biological replicates. ns, not significant, *p < 0.05, **p = 0.0043, ***p < 0.001, ****p < 0.0001, one-way ANOVA with Tukey's correction. (**L**) Quantification of AIA cell body length in wild-type, *ctbp-1,* and *ctbp-1; nEx2351* worms at L4 and day 1 adult (with or without heat shock) stages. Mean ± SEM. n ≥ 30 AIAs scored per strain per stage. ns, not significant, *p < 0.05, ****p < 0.0001, one-way ANOVA with Tukey's correction. The *ctbp-1* allele used for all panels of this figure was *n4784*. All strains contain *nIs840[P_{gcy-28.d}::gfp]*, and all strains other than 'Wild type' contain *nIs348[P_{ceh-28}::mCherry]* (not shown in images).

*Figure 3 continued on next page*

*Figure 3 continued*

Images are oriented such that left corresponds to anterior, top to dorsal.

The online version of this article includes the following source data and figure supplement(s) for figure 3:

**Source data 1.** Source data for *Figure 3* and supplements.

**Figure supplement 1.** Loss of *ctbp-1* results in a disruption of AIA morphology but not AIA size.

**Figure supplement 2.** Heat shock does not affect AIA morphology.

AIAs – adaptation to the volatile odor 2-butanone (*Cho et al., 2016*) – reasoning that if AIA function is disrupted in *ctbp-1* mutants, there should be a reduction of adaptation (and thus greater attraction) to butanone in *ctbp-1* worms relative to wild-type worms.

Consistent with previous studies (*Cho et al., 2016*), we found that worms that had been briefly starved with 90 minutes of food deprivation and had no prior experience with butanone (so-called 'naïve' worms) were generally attracted to the odor, while worms that were briefly starved in the presence of butanone ('conditioned' worms) adapted to the odor and exhibited mild repulsion to it (*Figure 4A–E*). We next compared wild-type and *ctbp-1* mutant worms for their ability to adapt to butanone. We found that while L1 *ctbp-1* worms showed an ability to adapt to butanone roughly similar to that of their wild-type counterparts, conditioned L4 *ctbp-1* mutants displayed a significant decrease in repulsion from butanone relative to wild-type L4 animals, indicating a decrease in their ability to adapt to the odor (*Figure 4B–E*). As a control, we assayed a strain carrying a transgenic construct that genetically ablates the AIA neurons, JN580. As expected, JN580 worms displayed decreased butanone adaptation at both the L1 and L4 larval stages. Thus, *ctbp-1* mutant worms displayed a defect in butanone adaptation similar to that of an AIA-ablated strain and did so only at a later larval stage, suggesting a potential loss of AIA function in L4 *ctbp-1* mutants. However, while *ctbp-1* mutant L4s exhibited weaker butanone adaptation than their wild-type counterparts, this defect was not as severe as that of JN580 L4s, indicating that *ctbp-1* mutant AIAs might retain some function. Additionally, the lack of a butanone adaptation defect in L1 *ctbp-1* mutants similar to that of L1 JN580 worms further suggests that loss of *ctbp-1* does not disrupt early AIA function and shows that *ctbp-1* is not required for the establishment of functional AIA neurons.

We next asked if *ctbp-1* can act cell-autonomously in the AIAs and in older worms to regulate butanone adaptation. We assayed *ctbp-1* mutants carrying the AIA-specific rescue construct *nIs743[P_{AIA}::ctbp-1(+)]* for butanone adaptation (*Figure 4B–E*) and found that AIA-specific restoration of *ctbp-1* rescued butanone adaption of conditioned *ctbp-1* mutant L4s to near wild-type levels (*Figure 4E*). We conclude that the butanone adaptation defect of *ctbp-1* mutants is caused by a disruption of AIA function and that *ctbp-1* can act cell-autonomously to regulate this AIA function. Next, we assayed *ctbp-1* mutants carrying the heat shock-inducible *ctbp-1(+)* rescue construct *nEx2351[P_{hsp}::ctbp-1(+)]* for butanone adaptation. We found that restoration of *ctbp-1* by heat shock at the L4 larval stage rescued the butanone adaptation defect in day one adults, indicating that *ctbp-1* can act in L4-to-day 1 adult worms to maintain proper AIA function after the initial establishment of the AIA cell identity (*Figure 4F–G*). Taken together, these data establish that loss of *ctbp-1* disrupts the function of the AIA neurons and that *ctbp-1* can act cell-autonomously and in L4-to-day 1 adult worms to maintain AIA function.

While conducting these assays, we observed that naïve *ctbp-1* mutant worms displayed a mildly weaker attraction to butanone than did their wild-type counterparts at both the L1 and L4 larval stages (*Figure 4B and D*). AIA-specific rescue of *ctbp-1* did not rescue this mild chemotaxis defect – naïve *ctbp-1* mutants carrying the *P_{AIA}::ctbp-1(+)* construct still displayed weaker butanone attraction than wild-type worms (*Figure 4B and D*). We suggest that this defect in attraction to butanone is not a consequence of dysfunction of the AIAs but rather of some other cell(s) involved in butanone chemotaxis. Consistent with this hypothesis, we found that *ctbp-1* mutants were defective in chemotaxis to the volatile odor isoamyl alcohol but were not defective in the response to diacetyl (*Figure 4—figure supplement 1A-B*) while AIA-ablated strain JN580 animals were not defective in either response (*Figure 4—figure supplement 1A-B*). These observations indicate that *ctbp-1* mutant worms have a broader defect in chemotaxis caused by the disruption of the function of cells other than the AIAs. Because our primary focus has been on how *ctbp-1* functions to maintaining the AIA cell identity, we did not attempt to identify the other cells with functions perturbed by the loss of *ctbp-1*.

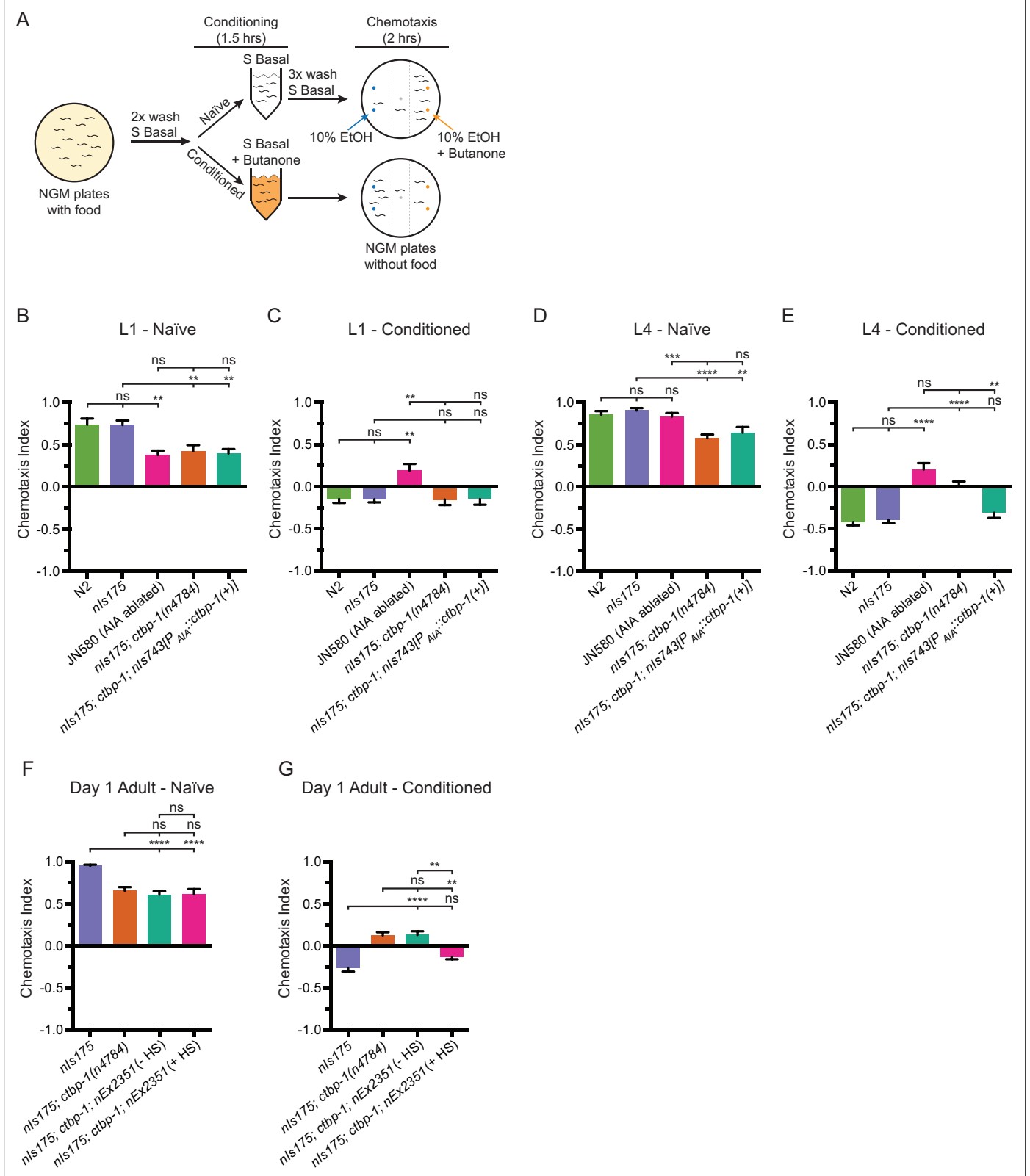

**Figure 4.** Loss of *ctbp-1* results in a disruption of AIA function in L4-to-day 1 adult worms. (**A**) Schematic of the butanone adaptation assay. L1 or L4 worms from synchronized populations were washed off plates with S Basal, washed with S Basal, split into naïve and conditioned populations, incubated in S Basal with or without 2-butanone for 1.5 hrs, washed again with S Basal, allowed to chemotax for 2 hrs on unseeded plates containing two 1 µl spots of 10% ethanol (blue dots) and 2-butanone diluted in 10% ethanol (orange dots), and then scored. (**B–E**) Chemotaxis indices of (**B,D**) naïve or

*Figure 4 continued on next page*

Figure 4 continued

(**C,E**) conditioned wild-type (N2 and *nIs175*), AIA-ablated (JN580), *nIs175; ctbp-1(n4784)*, and *nIs175; ctbp-1* mutants containing a transgene driving expression of wild-type *ctbp-1* under an AIA-specific promoter (*nIs743[P_{gcy-28.d}::ctbp-1(+)]*) at the (**B–C**) L1 or (**D–E**) L4 larval stage. Mean ± SEM. n ≥ 6 assays per condition, ≥ 50 worms per assay. ns, not significant, **p < 0.01, ***p = 0.0005, ****p < 0.0001, one-way ANOVA with Tukey's correction. (**F–G**) Chemotaxis indices of (**F**) naïve or (**G**) conditioned *nIs175*, *nIs175; ctbp-1*, and *nIs175; ctbp-1* mutants carrying the heat-shock-inducible transgene *nEx2351[P_{hsp-16.2}::ctbp-1(+); P_{hsp-16.41}::ctbp-1(+)]* with or without heat shock (HS) at the day 1 adult stage. Mean ± SEM. n ≥ 5 assays per condition, ≥ 50 worms per assay. ns, not significant, **p < 0.01, ****p < 0.0001, one-way ANOVA with Tukey's correction. The *ctbp-1* allele used for all panels of this figure was *n4784*.

The online version of this article includes the following source data and figure supplement(s) for figure 4:

**Source data 1.** Source data for *Figure 4* and supplements.

**Figure supplement 1.** *ctbp-1* mutants display a non-AIA-dependent chemotaxis defect.

## *ctbp-1* mutant AIAs have additional defects in gene expression

To better characterize the genetic changes occurring in mutant AIAs, we performed a single, exploratory single-cell RNA-Sequencing (scRNA-Seq) experiment comparing wild-type and *ctbp-1* mutant worms. We sequenced RNA from the neurons of wild-type and *ctbp-1* L4 worms and processed the resulting data using the 10X CellRanger pipeline to identify presumptive AIA neurons based on the expression of several AIA markers (*gcy-28*, *ins-1*, *cho-1*) shown above to be expressed in both wild-type and *ctbp-1* mutant AIAs (*Figure 2B*). Confirming that these data captured changes in the AIA transcriptional profiles, we found that *ctbp-1* mutant AIAs showed high levels of expression of *ceh-28*, while wild-type AIAs showed no detectable *ceh-28* expression (*Figure 5—figure supplement 1*).

We analyzed AIA transcriptional profiles to identify genes that appeared to be either expressed in *ctbp-1* mutant AIAs and not expressed in wild-type AIAs (similar to *ceh-28*) or expressed in wild-type AIAs but not expressed in *ctbp-1* AIAs. To confirm candidate genes, we crossed existing reporters for those genes to *ctbp-1* mutants or, in cases for which reporters were not readily available, generated our own transgenic constructs. We identified and confirmed one gene that, similar to *ceh-28*, was not expressed in wild-type AIAs but was misexpressed in *ctbp-1* mutant AIAs: *acbp-6*, which is predicted to encode an acyl-Coenzyme A binding protein (*Shaye and Greenwald, 2011*; *Figure 5A*; *Figure 5—figure supplement 1*). We also identified and confirmed two genes expressed in wild-type AIAs but not expressed in *ctbp-1* mutant AIAs: *sra-11*, which encodes a transmembrane serpentine receptor (*Troemel et al., 1995*); and *glr-2*, which encodes a glutamate receptor (*Brockie et al., 2001*; *Figure 5C and E*; *Figure 5—figure supplement 1*). We visualized the *acbp-6* reporter *nEx3081[P_{acbp-6}::gfp]*, the *sra-11* reporter *otIs123[P_{sra-11}::gfp]* and the *glr-2* reporter *ivEx138[P_{glr-2}::gfp]* in wild-type and *ctbp-1* L4 worms and confirmed that *acbp-6* was absent in wild-type AIAs but misexpressed in *ctbp-1* mutants (*Figure 5A–B*), while both *sra-11* and *glr-2* were consistently expressed in wild-type AIAs but not expressed in the AIAs of *ctbp-1* mutants (*Figure 5C–F*). We also visualized these reporters in L1 wild-type and *ctbp-1* worms and found that both *P_{acbp-6}::gfp* and *P_{sra-11}::gfp* displayed a time-dependence to their expression similar to that of *P_{ceh-28}::gfp* − *P_{acbp-6}::gfp* was rarely detectible in the AIAs of either wild-type or *ctbp-1* AIAs at the L1 stage but was consistently expressed in *ctbp-1* mutant L4 AIAs (*Figure 5A–B*), while *P_{sra-11}::gfp* was rarely detectible in the AIAs of either wild-type or *ctbp-1* mutant L1 worms but was expressed in the AIAs of most wild-type worms by the L4 stage while remaining off in the AIAs of most L4 *ctbp-1* mutants (*Figure 5C–D*). These observations suggest that, like *ceh-28* expression, *acbp-6* and *sra-11* expression is regulated by *ctbp-1* primarily in the AIAs of late-stage larvae and adults. By contrast, *glr-2* was expressed in wild-type but not *ctbp-1* AIAs in both L1 and L4 larvae (*Figure 5E–F*).

These data demonstrate that mutant AIAs fail to turn on and/or maintain the expression of genes characteristic of the adult AIA neuron (*sra-11* and *glr-2*) while misexpressing at least two genes uncharacteristic of AIA (*ceh-28* and *acbp-6*). That the majority of these abnormalities in AIA gene expression occurred long after the AIAs are generated during embryogenesis further supports the conclusion that *ctbp-1* does not act to establish the AIA cell identity.

Collectively, our findings concerning AIA gene expression, morphology and function demonstrate that *ctbp-1* acts to maintain the AIA cell identity, plays little to no role in the initial establishment of the AIA cell fate, and can act cell-autonomously and in older worms to maintain these aspects of the AIA identity.

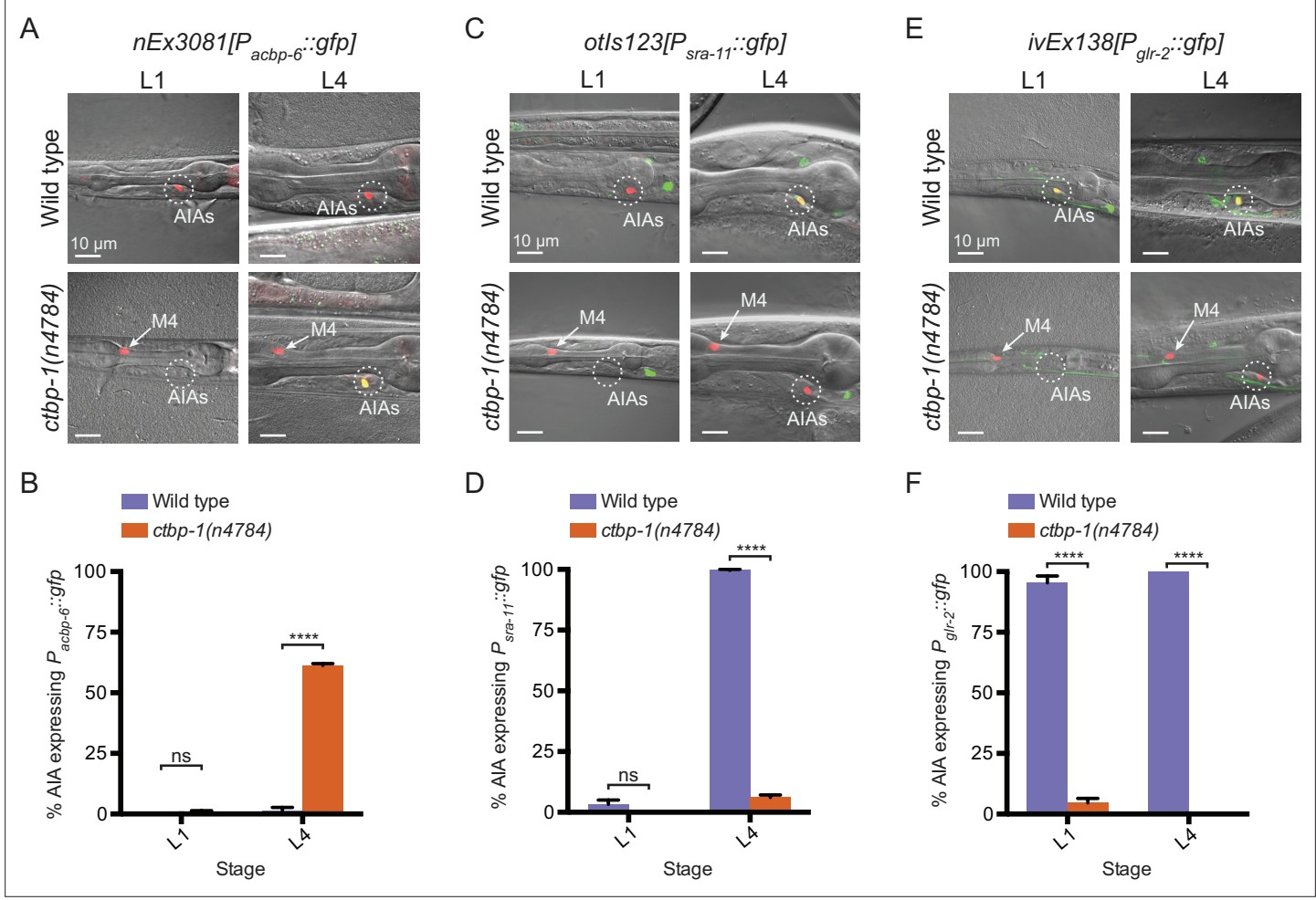

**Figure 5.** Loss of *ctbp-1* results in a disruption to normal AIA gene expression. (**A,C,E**) (**A**) *nEx3081[P_acbp-6::gfp]*, (**C**) *otls123[P_sra-11::gfp]*, or (**E**) *ivEx138[P_glr-2::gfp]* expression in wild-type (top) and *ctbp-1(n4784)* (bottom) worms at the L1 larval stage (left) and L4 larval stage (right). Wild-type strains contain *nls843[P_gcy-28.d::mCherry]*. *ctbp-1* mutant strains contain *nls348[P_ceh-28::mCherry]*. Arrow, M4 neuron. Circle, AIAs. Scale bar, 10 μm. (**B,D,F**) Percentage of wild-type and *ctbp-1(n4784)* expressing (**B**) *P_acbp-6::gfp*, (**D**) *P_sra-11::gfp*, or (**F**) *P_glr-2::gfp* in the AIA neurons at L1 and L4 larval stages. Wild-type strains contain *nls843[P_gcy-28.d::mCherry]*. *ctbp-1* mutant strains contain *nls348[P_ceh-28::mCherry]*. Mean ± SEM. n ≥ 50 worms per strain per stage, three biological replicates. ns, not significant, ****$p < 0.0001$, unpaired t-test.

The online version of this article includes the following source data and figure supplement(s) for figure 5:

**Source data 1.** Source data for *Figure 5* and supplements.

**Figure supplement 1.** Gene expression in wild-type and *ctbp-1* AIAs.

## Mutation of *egl-13* or *ttx-3* suppresses the *ctbp-1* mutant phenotype

To investigate how *ctbp-1* acts to maintain AIA cell identity, we performed a mutagenesis screen for suppression of *P_ceh-28::gfp* misexpression in the AIAs of L4 *ctbp-1* mutants (*Figure 6A*). Using a combination of Hawaiian SNP mapping (*Davis et al., 2005*) and whole-genome sequencing, we identified two genes as suppressors of the *ctbp-1* mutant phenotype: *egl-13*, which encodes a SOX family transcription factor; and *ttx-3*, which encodes a LIM homeobox transcription factor. *egl-13* has been shown to act in the establishment of the BAG and URX cell fates and in vulval development of *C. elegans* (*Gramstrup Petersen et al., 2013*; *Feng et al., 2013*), and its mammalian orthologs SOX5 and SOX6 act in neural fate determination (*Ji and Kim, 2016*; *Saleem et al., 2020*). We isolated three alleles of *egl-13* as *ctbp-1* suppressors: *n5937*, a mutation of the splice acceptor site at the beginning of the 6[th] exon of the *egl-13a* isoform resulting in a frameshift and early stop; *n6013*, a Q381ochre nonsense mutation toward the end of the *egl-13* transcript; and *n6313*, a 436-nucleotide deletion spanning the 7th and 8th exons of the *egl-13a* isoform (*Figure 6B*; *Figure 6—figure supplement*

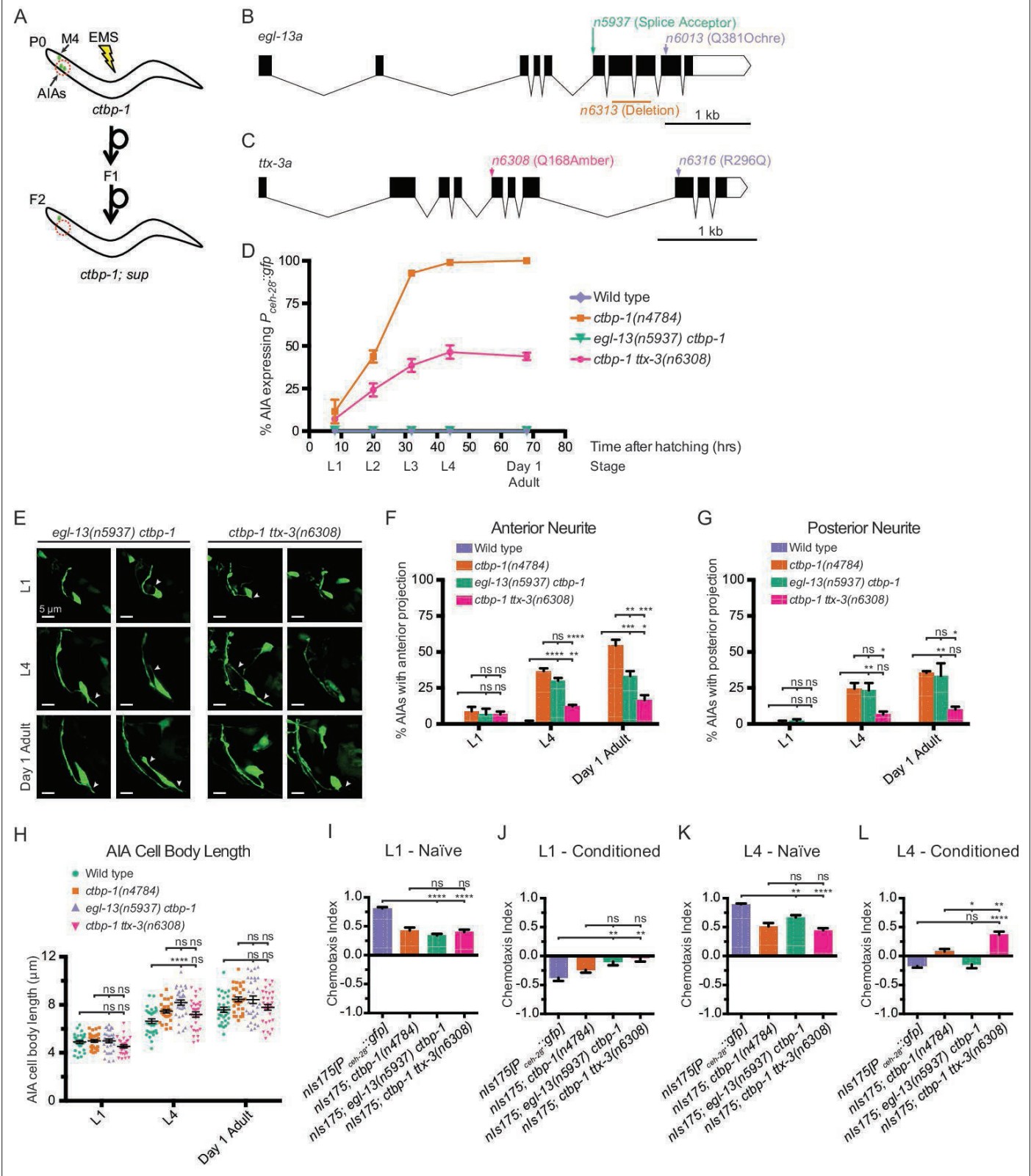

**Figure 6.** A suppressor screen reveals *egl-13* and *ttx-3* as *ctbp-1* genetic interactors. (**A**) Schematic of *ctbp-1* suppressor screen design. *ctbp-1* mutant worms carrying *nls175[P_ceh-28::gfp]* were mutagenized with ethyl methanesulfonate (EMS), and their F2 progeny were screened for continued *nls175* expression in M4 and loss of expression in the AIA neurons (red circle). (**B**) Gene diagram of the *egl-13a* isoform. Arrows (above), point mutations. Line (below), deletion. Scale bar (bottom right), 1 kb. (**C**) Gene diagram of the *ttx-3a* isoform. Arrows (above), point mutations. Scale bar, 1 kb. (**D**) Percentage

*Figure 6 continued on next page*

*Figure 6 continued*

of wild-type, *ctbp-1*, *egl-13(n5937) ctbp-1*, and *ctbp-1 ttx-3(n6308)* worms expressing *nIs175* in the AIA neurons over time. Time points correspond to the L1, L2, L3, L4 larval stages, and day 1 adult worms (indicated below X axis). All strains contain *nIs175[P_ceh-28::gfp]*. Mean ± SEM. n ≥ 100 worms per strain per stage, three biological replicates. (**E**) Two representative images of an AIA neuron in *egl-13 ctbp-1* or *ctbp-1 ttx-3* worms at L1 (top), L4 (middle), and day 1 adult (bottom) stages. Arrows, examples of ectopic neurites protruding from the AIA cell body. Image oriented such that left corresponds to anterior, top to dorsal. Scale bar, 5 µm. (**F–G**) Percentage of AIAs in wild-type, *ctbp-1*, *egl-13 ctbp-1* and *ctbp-1 ttx-3* worms at the L1, L4, and day 1 adult stages with an ectopic neurite protruding from the (**F**) anterior or (**G**) posterior of the AIA cell body. Mean ± SEM. n = 30 AIAs scored per strain per stage, three biological replicates. ns, not significant, *p < 0.05, **p < 0.01, ***p < 0.001, ****p < 0.0001, one-way ANOVA with Tukey's correction. (**H**) Quantification of AIA cell body length in wild-type, *ctbp-1*, *egl-13 ctbp-1*, and *ctbp-1 ttx-3* worms at the L1, L4, and day 1 adult stages. Mean ± SEM. n ≥ 30 AIAs scored per strain per stage. ns, not significant, ****p < 0.0001, one-way ANOVA with Tukey's correction. (**I–L**) Chemotaxis indices of (**I,K**) naïve or (**J,L**) conditioned wild-type, *ctbp-1*, *egl-13 ctbp-1*, and *ctbp-1 ttx-3* worms at the (**I–J**) L1 or (**K–L**) L4 larval stage. Mean ± SEM. n ≥ 5 assays per condition, ≥ 50 worms per assay. ns, not significant, *p = 0.0214, **p < 0.01, ****p < 0.0001, one-way ANOVA with Tukey's correction. The *ctbp-1* allele used for all panels of this figure was *n4784*. The *egl-13* allele used for all panels of this figure was *n5937*. The *ttx-3* allele used for all panels of this figure was *n6308*. All strains in (**E–H**) contain *nIs840[P_gcy-28.d::gfp]* and all strains in (**E–H**) other than 'Wild type' contain *nIs348[P_ceh-28::mCherry]* (not shown in images). All strains in (**D, I–L**) contain *nIs175[P_ceh-28::gfp]*.

The online version of this article includes the following source data and figure supplement(s) for figure 6:

**Source data 1.** Source data for *Figure 6* and supplements.

**Figure supplement 1.** Characterization of *egl-13* alleles isolated as *ctbp-1* suppressors.

**Figure supplement 2.** Characterization of *ttx-3* alleles isolated as *ctbp-1* suppressors.

---

*1A-B*). We generated and introduced a transgenic construct carrying a wild-type copy of *egl-13* under its native promoter into these mutant strains and found that this construct was capable of rescuing the suppression of *P_ceh-28::gfp* misexpression by all three *egl-13* alleles, demonstrating that loss of *egl-13* function suppresses this aspect of the *ctbp-1* mutant phenotype and suggesting that these alleles are likely loss-of-function alleles of *egl-13* (*Figure 6—figure supplement 1C*).

*ttx-3* is a LIM homeobox transcription factor characterized for its roles in thermotaxis behavior and in cell-fate specification of the AIA and AIY interneurons (*Altun-Gultekin et al., 2001*; *Zhang et al., 2014*). The TTX-3 mammalian ortholog LHX9 is involved in retinal cell-fate establishment (*Balasubramanian et al., 2014*; *Balasubramanian et al., 2018*). We isolated two alleles of *ttx-3*: the nonsense allele *n6308* and the missense allele *n6316* (*Figure 6C*; *Figure 6—figure supplement 2A-B*). We tested two additional *ttx-3* alleles, the splice acceptor allele *ks5* and the Q303amber nonsense allele *ot22*. Both suppressed *ctbp-1*-driven *nIs175* misexpression (*Figure 6—figure supplement 2B*). Suppression of *ctbp-1* by either of our isolated alleles (*n6308* or *n6316*) was rescued by introduction of a transgenic construct carrying a wild-type copy of *ttx-3* expressed under the control of its native promoter (*Figure 6—figure supplement 2C*), indicating that these alleles are likely loss-of-function alleles that reduce or eliminate *ttx-3* gene function.

We assayed the loss-of-function alleles *egl-13(n5937)* and *ttx-3(n6308)* for their ability to suppress *P_ceh-28::gfp* misexpression over the course of larval development and into adulthood of *ctbp-1* mutant worms (*Figure 6D*). *egl-13(n5937)* strongly suppressed *ctbp-1* at all stages, resulting in little to no misexpression of *P_ceh-28::gfp* in the AIAs of *egl-13 ctbp-1* double mutants at any larval stage or in day 1 adults. Suppression by *ttx-3(n6308)* was incompletely penetrant, showing a progressive increase in AIA gene misexpression that peaked at adulthood with approximately 45% of *ctbp-1 ttx-3* double-mutant animals displaying *P_ceh-28::gfp* expression in the AIAs.

We next asked if *egl-13(n5937)* or *ttx-3(n6308)* could suppress the AIA morphological and functional defects of *ctbp-1* mutants. To both test suppression of AIA morphological defects and confirm the presence of the AIA neurons in *egl-13 ctbp-1* and *ctbp-1 ttx-3* double mutants, we crossed the AIA morphology reporter *nIs840[P_gcy-28.d::gfp]* into these double mutants and scored AIA morphology in L1, L4 and day one adult worms (*Figure 6E–H*). *egl-13 ctbp-1* double mutant AIAs displayed a mild (though significant) reduction in the penetrance of ectopic anterior neurites only in adult worms and no significant change in the frequency of posterior neurites or AIA cell body length at any stage. *ctbp-1 ttx-3* double mutants displayed a significant decrease in the frequency of both ectopic anterior and posterior projections at the L4 and day one adult stages (though these double mutants still displayed a greater frequency of these defects than did their wild-type counterparts) and no significant difference in AIA cell body length at any stage tested. These data demonstrate that loss of *egl-13* has little consistent effect on the AIA morphological defects caused by a loss of *ctbp-1* activity,

suggesting that *ctbp-1* maintains AIA morphology primarily through *egl-13*-independent pathways. These data further indicate that the additional loss of *ttx-3* results in less severe morphological abnormalities than occurs in *ctbp-1* single mutants, suggesting that the manifestation of AIA morphological defects seen in *ctbp-1* single mutants likely rely in part on proper *ttx-3* activity.

We next assayed the ability of *egl-13(n5937)* and *ttx-3(n6308)* to suppress AIA functional defects. We tested *egl-13 ctbp-1* and *ctbp-1 ttx-3* double mutants for butanone adaptation and found that, at the L1 larval stage, these double mutant strains displayed a detectable response to butanone similar to *ctbp-1* single mutants (*Figure 6I–J*). At the L4 larval stage, mutation of *egl-13* strongly suppressed the *ctbp-1* mutant defect in butanone adaptation, causing near wild-type levels of repulsion in conditioned worms (*Figure 6K–L*). By contrast, at the L4 stage mutation of *ttx-3* failed to suppress AIA functional defects and instead showed an even greater defect in butanone adaption than did *ctbp-1* single mutants (*Figure 6K–L*). These results indicate that loss of *egl-13* activity suppressed AIA functional defects of *ctbp-1* mutant worms and suggest that *ctbp-1* maintains at least this aspect of AIA cellular function primarily through an *egl-13*-dependent pathway, while loss of *ttx-3* exacerbates the AIA functional defect seen in *ctbp-1* single mutants.

From these data, we conclude that *ctbp-1* maintains AIA function and at least some aspects of AIA gene expression through *egl-13* and that disruptions to AIA morphology appears to be controlled independently of *egl-13* function. We further conclude that loss of *ttx-3* similarly restores at least some aspects of AIA gene expression as well as partially suppresses AIA morphological defects seen in *ctbp-1* single mutants, while AIA function seems to be further perturbed in the absence of *ttx-3*.

Mutation of *ctbp-1* does not affect *ttx-3* expression in the AIAs (*Figure 2B*), suggesting that the AIA identity defects that characterize *ctbp-1* single mutants are not a consequence of a change in *ttx-3* expression. Given *ttx-3*'s known requirement as an AIA terminal selector (*Zhang et al., 2014*), we speculate that mutation of *ttx-3* appears to suppress AIA cell-identity maintenance defects caused by a loss of *ctbp-1* by perturbing AIA cell-identity establishment. In other words, failure to properly establish the AIA cell identity, as likely occurs in *ttx-3* mutants, masks defects caused by the loss of the maintenance of that cell identity, as occurs in *ctbp-1* mutants. While these mutants might offer insights into the interplay between cell-identity establishment and maintenance, our primary focus was on the mechanisms of cell-identity maintenance and we focused our further efforts on characterizing the EGL-13 – CTBP-1 relationship and how these two proteins act to maintain the AIA cell identity.

## EGL-13 can function cell-autonomously in the AIAs and likely physically interacts with CTBP-1

To determine if, like *ctbp-1*, *egl-13* can act cell-autonomously in the AIAs, we generated a transgenic construct that drives expression of a wild-type copy of *egl-13* in the AIAs ($nEx3055[P_{gcy-28.d}::egl-13(+)]$). Introduction of this construct to *egl-13(n5937) ctbp-1* double mutants rescued the *egl-13* suppression of $P_{ceh-28}::gfp$ misexpression in the AIAs, indicating that *egl-13* can function cell-autonomously (*Figure 7—figure supplement 1A-B*). We further tested *egl-13* expression using a GFP transcriptional reporter and found that *egl-13* was expressed in a number of cells, including the AIAs, of wild-type worms (*Figure 7—figure supplement 2A*), although expression in the AIAs appeared to dissipate over the course of larval development (*Figure 7—figure supplement 2C*). We also tested the effect of mutation of *ctbp-1* on *egl-13* expression and found no significant difference between wild-type and *ctbp-1* worms at any stage tested, indicating that *ctbp-1* does not appear to regulate *egl-13* expression in the AIAs (*Figure 7—figure supplement 2B-C*). These results suggest that, in the absence of *ctbp-1* function, ectopic *egl-13* activity in the AIAs drives *ceh-28* misexpression, and thus that *ctbp-1* likely normally acts to repress *egl-13* activity in the AIAs. These findings further raise the possibility that EGL-13 and CTBP-1 might be interacting within the AIAs to coordinate maintenance of the AIA cell identity.

The mammalian CtBPs (CtBP1 and CtBP2) bind PXDLS-like motifs on a number of diverse transcription factors to target specific genetic loci for silencing (*Chinnadurai, 2003*; *Shi et al., 2003*; *Stankiewicz et al., 2014*). The mammalian ortholog of EGL-13 (SOX6) interacts with the mammalian ortholog of CTBP-1 (CtBP2) through a PLNLS motif located in SOX6 to repress *Fgf-3* expression in the developing mouse auditory otic vesicle (*Murakami et al., 2001*). This motif is 100% conserved in *C. elegans* EGL-13 and, since both CTBP-1 and EGL-13 can function cell-autonomously in the AIAs, we hypothesized that CTBP-1 and EGL-13 might physically interact. We performed a yeast 2-hybrid assay

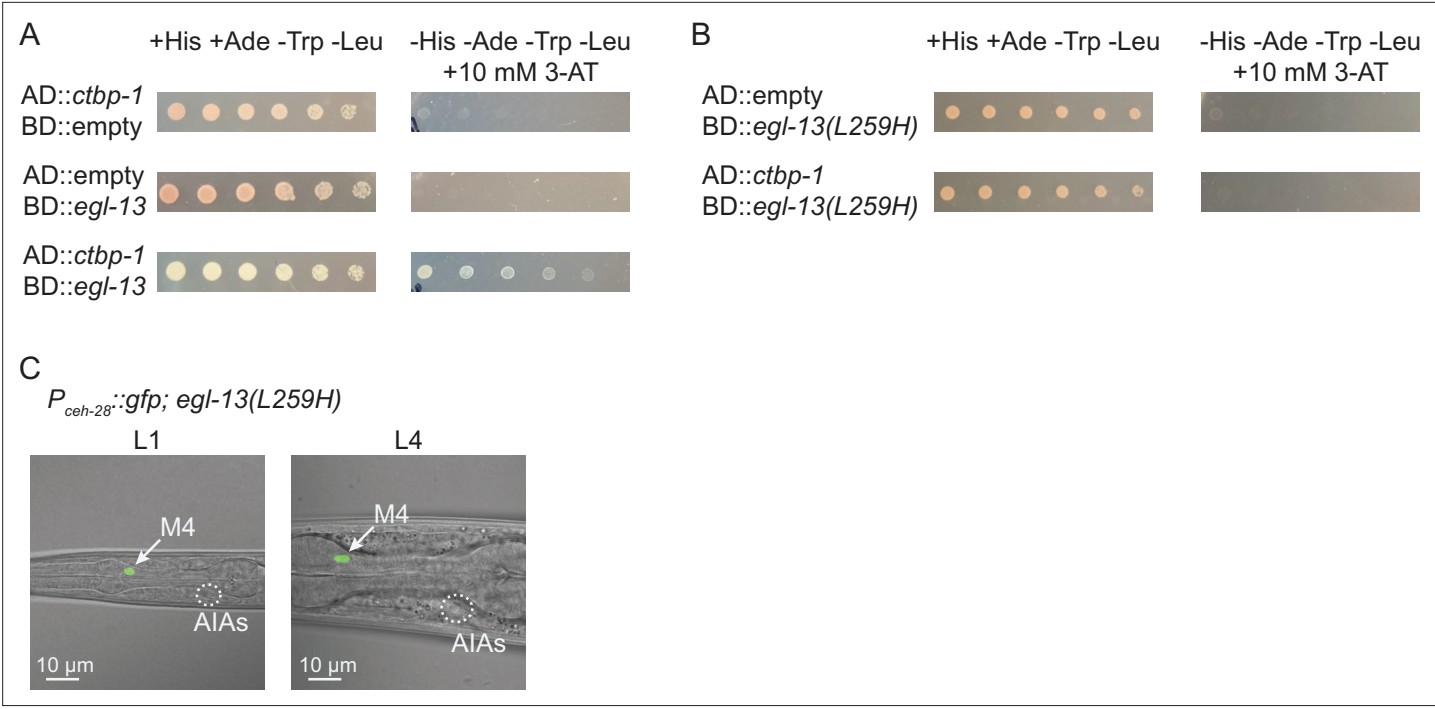

**Figure 7.** CTBP-1 can physically bind EGL-13 through EGL-13's conserved PLNLS domain. (**A**) Serial dilution of yeast colonies carrying plasmids containing the Gal4 Activating Domain (AD) and Gal4 DNA-Binding Domain (BD) fused to *ctbp-1a* cDNA, *egl-13a* cDNA, or neither ('empty'). Strains carrying both domain-containing plasmids grow on+ His + Ade -Trp -Leu plates (left). Strains in which the proteins interact grow on -His -Ade -Trp -Leu + 10 mM 3-AT plates (right). (**B**) Serial dilution of yeast colonies carrying plasmids containing the Gal4 Activating Domain (AD) and Gal4 DNA-Binding Domain (BD) fused to *ctbp-1a* cDNA, *egl-13a* cDNA with amino acids 256–260 mutated from PLNLS to PLNHS ('*egl-13(L259H)*'), or neither ('empty'). (**C**) Representative images of (left) L1 and (right) L4 *egl-13(n6675)* mutants in which amino acid 259 was mutated from Leu to His ('*egl-13(L259H)*') displaying *nIs175[P_{ceh-28}::gfp]* expression. Arrow, M4 neuron. Circle, AIAs. Scale bar, 10 μm. Images are oriented such that left corresponds to anterior, top to dorsal. Quantification of reporter expression in *Figure 7—figure supplement 3*.

The online version of this article includes the following source data and figure supplement(s) for figure 7:

**Source data 1.** Source data for *Figure 7* and supplements.

**Figure supplement 1.** EGL-13 functions cell-autonomously to regulate AIA gene expression.

**Figure supplement 2.** *P_{egl-13}::gfp* expression in wild-type and *ctbp-1* mutant worms.

**Figure supplement 3.** Quantification of *P_{ceh-28}::gfp* expression in *egl-13(n6675)* mutants.

to test this hypothesis and found that CTBP-1 and EGL-13 are indeed able to physically interact in this assay (*Figure 7A*). These previous studies have further shown that the SOX6 – CtBP2 interaction can be disrupted by a leucine-to-histidine mutation of the second leucine in the PLNLS motif (to PLNHS), resulting in a mutant SOX6 protein that is unable to interact with CtBP2 (*Murakami et al., 2001*). We generated a mutant EGL-13 variant bearing an L259H mutation, disrupting this motif, and found that this mutation disrupted its interaction with CTBP-1 in a yeast 2-hybrid assay (*Figure 7B*), suggesting that EGL-13's PLNLS motif is critical for the EGL-13 – CTBP-1 interaction.

We hypothesized that disruption of this interaction in an otherwise wild-type worm might be sufficient to drive misexpression of *P_{ceh-28}::gfp* in the AIAs. To test this hypothesis, we used CRISPR (*Dickinson and Goldstein, 2016*) to generate the L259H mutation of *egl-13* in a wild-type background and assayed *P_{ceh-28}::gfp* expression in the AIAs. We found that this mutation did not induce reporter misexpression at either the L1 or L4 larval stages in any of the worms scored (*Figure 7C*; quantified in *Figure 7—figure supplement 3*), indicating that disruption of the EGL-13 – CTBP-1 interaction alone in not sufficient to induce misexpression of *ceh-28* in the AIAs.

### *egl-13* regulates some aspects of AIA gene expression

We next asked if mutation of *egl-13* could suppress other *ctbp-1* mutant AIA gene expression defects besides that of *ceh-28*. We crossed in *acbp-6*, *sra-11* and *glr-2* reporters to *egl-13 ctbp-1* double

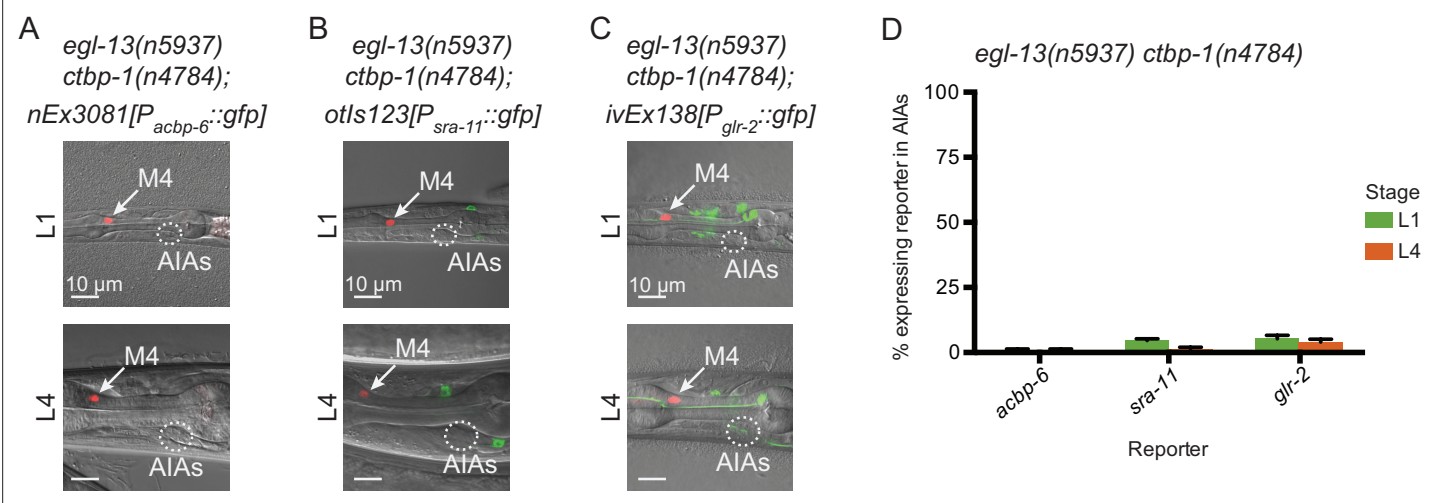

**Figure 8.** EGL-13 controls aspects of AIA gene expression. (**A–C**) Expression of markers for AIA misexpressed genes (**A**) *nEx3081[P_acbp-6_::gfp]*, (**B**) *otIs123[P_sra-11_::gfp]*, or (**C**) *ivEx138[P_glr-2_::gfp]* in *egl-13(n5937) ctbp-1(n4784)* double mutants at the (top) L1 and (bottom) L4 larval stages. Arrow, M4 neuron. Circle, AIAs. Scale bar, 10 μm. (**D**) Percentage of *egl-13(n5937) ctbp-1(n4784)* double mutants expressing the indicated reporter in the AIA neurons at the L1 and L4 larval stages. Mean ± SEM. n ≥ 50 worms scored per strain, three biological replicates. The *ctbp-1* allele used for all panels of this figure was *n4784*. All strains in **A-D** contain *nIs348[P_ceh-28_::mCherry]*. Images are oriented such that left corresponds to anterior, top to dorsal.

The online version of this article includes the following source data for figure 8:

**Source data 1.** Source data for *Figure 8*.

mutants and visualized reporter expression at the L1 and L4 larval stages. We found that mutation of *egl-13* suppressed *P_acbp-6_::gfp* misexpression in the AIAs (*Figure 8A and D*, compare to *Figure 5*), just as *egl-13* mutation suppressed *P_ceh-28_::gfp* misexpression. By contrast, mutation of *egl-13* had no effect on the loss of *P_sra-11_::gfp* or *P_glr-2_::gfp* expression in *ctbp-1* mutants (*Figure 8B–D*). We speculated that EGL-13 might directly regulate expression of *ceh-28* or *acbp-6*. To test this hypothesis, we examined the *ceh-28* and *acbp-6* promoter regions for possible EGL-13 binding sites. We failed to identify any promising candidates, suggesting that regulation of these genes by EGL-13 is likely indirect. These results demonstrate that some, though not all, of the AIA gene expression defects seen in *ctbp-1* mutants are regulated through *egl-13*.

### *egl-13* regulates AIA function through control of *ceh-28* expression

As *egl-13* is required for misexpression of both *ceh-28* and *acbp-6* as well as for disruption of AIA function in *ctbp-1* mutants, we hypothesized that misexpressed *ceh-28* or *acbp-6* might be contributing to the observed AIA functional defect in *ctbp-1* mutants. If so, we expected that mutations that eliminated the functions of these ectopically expressed genes should restore AIA function in *ctbp-1* mutants. To test this hypothesis, we crossed mutant alleles of *ceh-28 (cu11)* or *acbp-6 (tm2995)* (both deletion alleles spanning greater than half their respective genes) to *ctbp-1(n4784)* mutants and assayed the resulting double mutants for butanone adaptation in L1 and L4 worms. We found that *acbp-6; ctbp-1* double mutants were nearly identical to both naïve and conditioned *ctbp-1* single mutants at both the L1 and L4 larval stages (*Figure 9A–D*), indicating that misexpressed *acbp-6* is likely not responsible for the observed AIA functional defect. Conditioned *ctbp-1 ceh-28* double mutants appeared similar to both the wild type and *ctbp-1* single mutants at the L1 stage (*Figure 9E–F*). These double mutants display a minor, although not statistically significant, difference in adaptation at the L4 larval stage when compared to either wild-type or *ctbp-1* animals (*Figure 9G–H*).

These data raise the possibility that *ceh-28* overexpression in the AIAs can perturb AIA function. To test this hypothesis, we generated a transgenic strain (*nIs753[P_gcy-28.d_::ceh-28]*) that overexpresses *ceh-28* specifically in the AIA neurons and tested this strain for butanone adaptation. We found that overexpression of *ceh-28* in the AIAs resulted in a minor decrease in butanone attraction in naïve worms at both the L1 and L4 stage, although in both cases not more severe than the defect seen in *ctbp-1* mutants (*Figure 9I and K*). However, at both the L1 and L4 larval stages, AIA-specific *ceh-28*

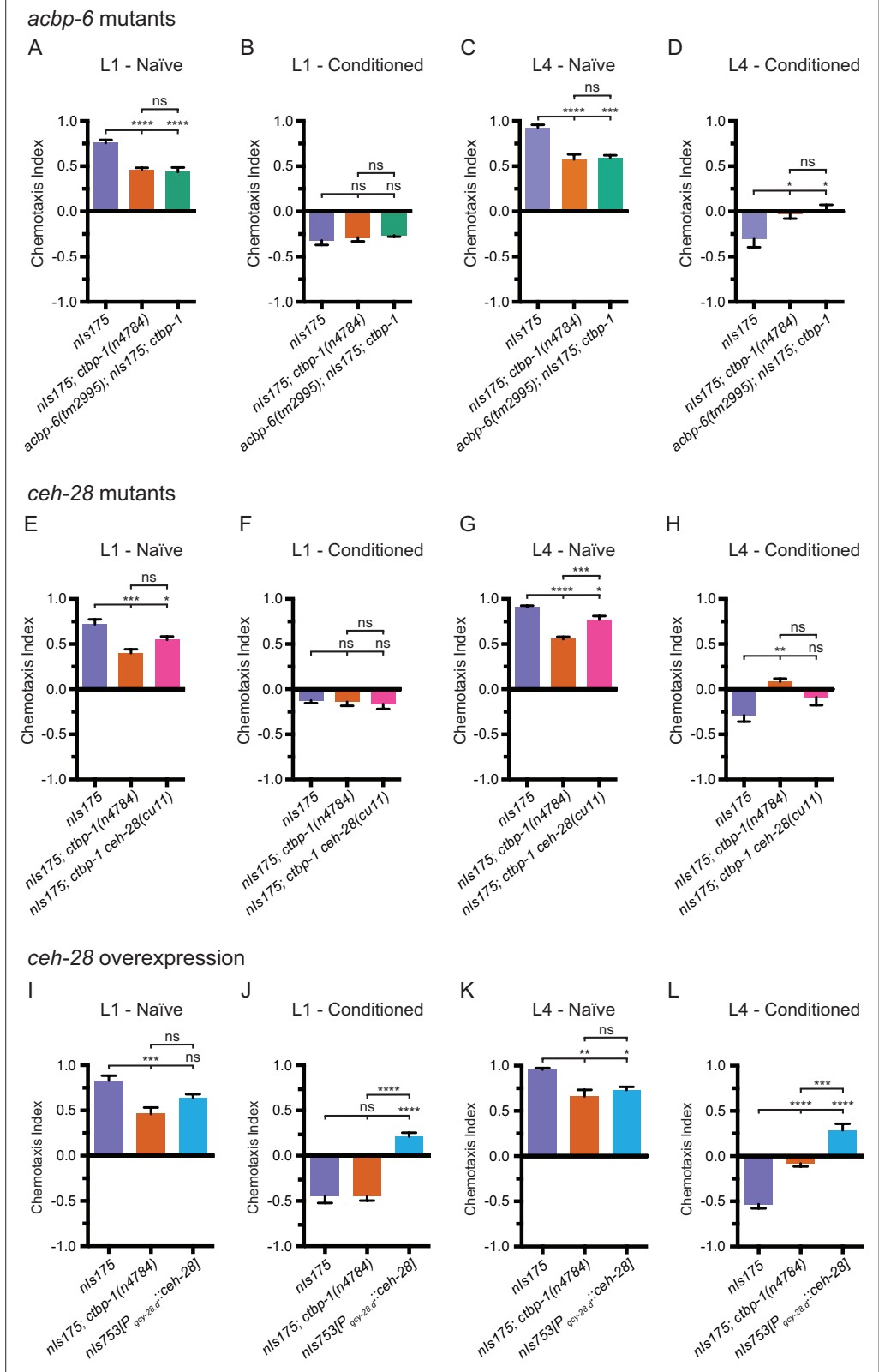

**Figure 9.** EGL-13 disrupts AIA function partially through driving misexpression of *ceh-28* in *ctbp-1* mutants. (**A–D**) Chemotaxis indices of (**A,C**) naïve or (**B,D**) conditioned wild-type (*nIs175*), *nIs175; ctbp-1(n4784)*, and *acbp-6(tm2995); nIs175; ctbp-1* mutants at the (**A–B**) L1 or (**C–D**) L4 larval stage. Mean ± SEM. n ≥ 6 assays per condition, ≥ 50 worms per assay. ns, not significant, *p < 0.05, ***p = 0.0003, ****p < 0.0001, one-way ANOVA with Tukey's

*Figure 9 continued on next page*

*Figure 9 continued*

correction. (**E–H**) Chemotaxis indices of (**E,G**) naïve or (**F,H**) conditioned wild-type (*nIs175*), *nIs175; ctbp-1(n4784)*, and *nIs175; ctbp-1 ceh-28(cu11)* mutants at the (**E–F**) L1 or (**G–H**) L4 larval stage. Mean ± SEM. n ≥ 6 assays per condition, ≥ 50 worms per assay. ns, not significant, *p < 0.05, **p = 0.0031, ***p < 0.001, ****p < 0.0001 one-way ANOVA with Tukey's correction. (**I–L**) Chemotaxis indices of (**I,K**) naïve or (**J,L**) conditioned wild-type (*nIs175*), *nIs175; ctbp-1(n4784)*, and *nIs753[P_{gcy-28.d}::ceh-28(+)]* at the (**I–J**) L1 or (**K–L**) L4 larval stage. Mean ± SEM. n ≥ 6 assays per condition, ≥ 50 worms per assay. ns, not significant, *p = 0.0176, **p = 0.0018, ***p < 0.001, ****p < 0.0001, one-way ANOVA with Tukey's correction. The *ctbp-1* allele used for all panels of this figure was *n4784*.

The online version of this article includes the following source data for figure 9:

**Source data 1.** Source Data for *Figure 9*.

overexpression resulted in a significant increase in attraction to butanone in conditioned worms (and thus a decrease in adaptation) (*Figure 9J and L*). These results demonstrate that *ceh-28* overexpression in the AIAs is sufficient to perturb AIA function and suggest that misregulation of *ceh-28* expression in the AIAs of *ctbp-1* mutants might be partially responsible for the disruption of AIA function seen in these mutants. Additionally, the striking difference at the L1 stage between *ctbp-1* mutants and worms that overexpress *ceh-28* in the AIAs supports the idea that early larval stage AIAs are not dysfunctional in *ctbp-1* mutants.

Collectively, these data suggest that overexpression of *ceh-28*, caused by a loss of *ctbp-1* and likely driven by ectopic *egl-13* activity, might partially account for the defect in butanone adaptation seen in L4 and day 1 adult *ctbp-1* mutants and that removal of *egl-13* in part restores AIA function by eliminating *ceh-28* misexpression. We propose that *ctbp-1* functions to maintain aspects of the AIA cell identity by preventing *egl-13* from promoting *ceh-28* expression and that *ceh-28* misexpression can perturb proper AIA function. These results also indicate that *ceh-28* misexpression alone is not solely responsible for the observed AIA functional defect, suggesting that the regulation of other, as-of-yet unidentified genes controlled by *ctbp-1* (and potentially *egl-13*) also contribute to the maintenance of the AIA cell identity.

## Discussion

We have shown that the *ctbp-1* transcriptional corepressor gene is required to maintain AIA cell identity and that *ctbp-1* negatively and selectively regulates the function of the *egl-13* transcription factor gene. We suggest that the CTBP-1 protein functions as a transcriptional corepressor to selectively regulate the transcriptional output (either directly or indirectly) of the EGL-13 protein. c*tbp-1* mutant AIAs undergo a progressive decline in their initially wild-type gene-expression pattern, morphology and function. *ctbp-1* can act cell-autonomously and is able to act in older animals to maintain these aspects of the AIA identity. We conclude that CTBP-1 functions to maintain AIA cell identity and speculate that other transcriptional corepressors similarly function in the maintenance of specific cell identities and do so by silencing undesired gene expression through repression of transcriptional activators, such as EGL-13. Such a mechanism could explain how the breadth of transcriptional activation by terminal selectors can be fine-tuned in a coordinated fashion to fit the requirements of specific cell types, with selective transcriptional silencing providing a crucial aspect of proper cell-identity maintenance.

### CTBP-1 physically interacts with EGL-13 to maintain the AIA cell identity

Our findings (*Figure 10A*) suggest that CTBP-1 interacts with EGL-13 to regulate EGL-13 activity as part of AIA cell-identity maintenance. We propose a model (*Figure 10B–D*) in which CTBP-1 physically interacts with EGL-13 to target specific genetic loci for silencing as an aspect of normal AIA cell-identity maintenance. Following the establishment of the AIA cell fate, for which CTBP-1 is not required, CTBP-1 binds EGL-13, recruiting CTBP-1 to EGL-13 DNA binding sites. CTBP-1 then silences surrounding genetic loci, resulting in the repression of specific target genes (*Figure 10B*). This repression is necessary for proper maintenance of the AIA cell identity, and when disrupted, as in *ctbp-1* mutants, CTBP-1 binding partners, such as EGL-13, inappropriately act as transcriptional activators in the AIAs, resulting in disruption of AIA gene expression, morphology and function (*Figure 10C*). In the

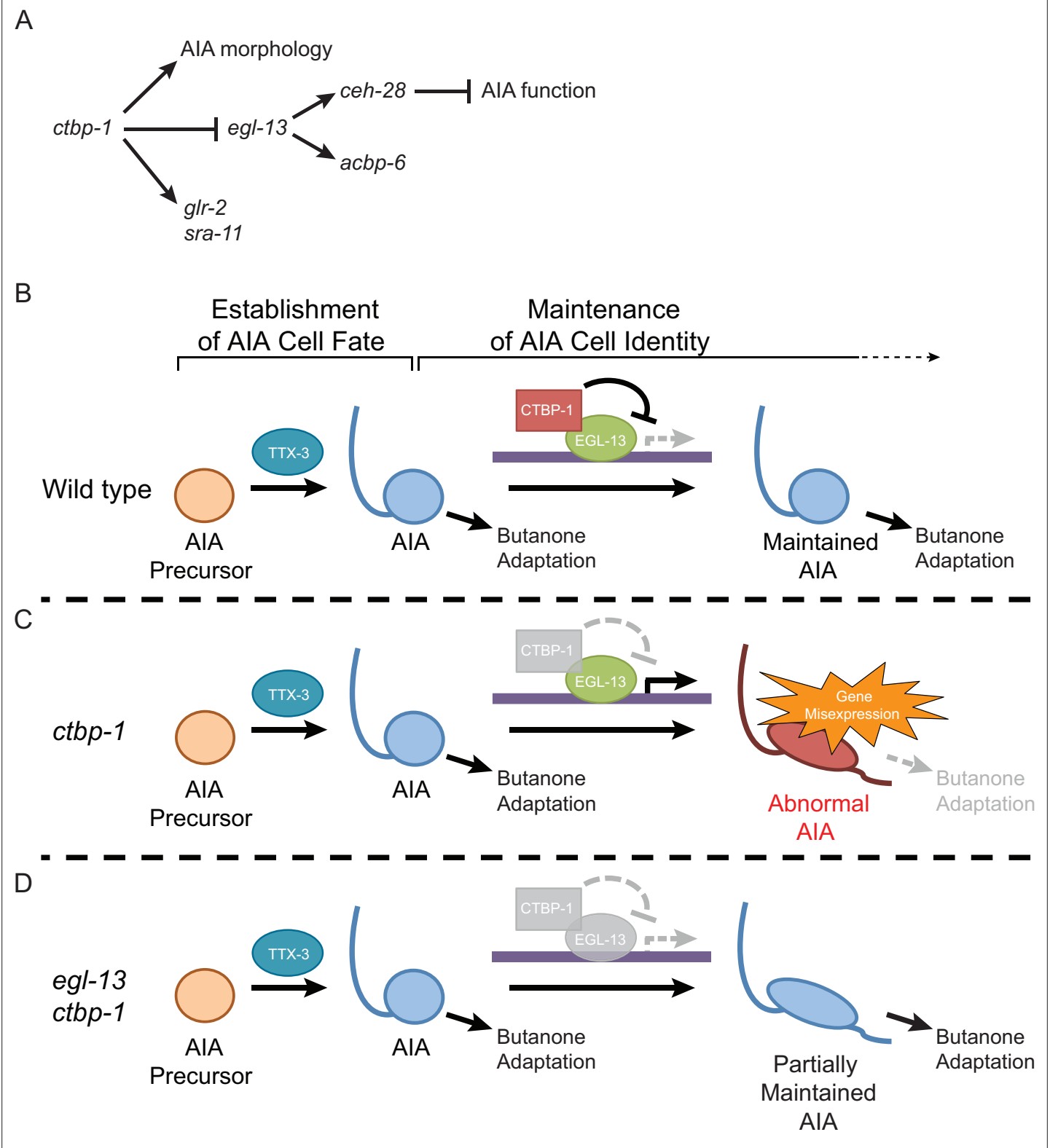

**Figure 10.** Model for the maintenance of the AIA cell identity by *ctbp-1*. (**A**) The genetic pathway in which *ctbp-1* promotes AIA morphology and *glr-2* and *sra-11* expression. *ctbp-1* also inhibits *egl-13*, thereby repressing expression of *ceh-28* and *acbp-6* in the AIAs and promoting proper AIA function (**B–D**) Model for how CTBP-1 maintains the AIA cell identity. (**B**) We propose that CTBP-1 acts in the maintenance but not establishment of the AIA cell identity, and does so by targeting specific genetic loci for regulation through physical interaction with transcription factors such as EGL-13. TTX-3 is required for the establishment, but not the maintenance, of the AIA identity. (**C**) In the absence of CTBP-1, EGL-13 and other CTBP-1 targets drive

*Figure 10 continued on next page*

*Figure 10 continued*

expression at multiple genetic loci, resulting in changes to the gene expression, morphology and function (as assessed by butanone adaptation) of the AIAs. (**D**) When EGL-13 activity is also removed, gene expression and cellular function are no longer perturbed, while normal morphology is not restored, resulting in a 'Partially Maintained AIA'.

absence of such CTBP-1 interactors, as in *egl-13 ctbp-1* double mutants, aberrant transcription is not activated and some of the defects in AIA maintenance are avoided (*Figure 10D*).

## TTX-3 helps establishing the AIA cell identity but does not maintain it with CTBP-1

We speculate that mutations in *ttx-3* appear to suppress the *ctbp-1* mutant phenotype of abnormal AIA cell-identity maintenance because *ttx-3* mutations are epistatic to *ctbp-1* mutations: in *ttx-3 ctbp-1* double mutants the AIA cell fate is not established, resulting in a lack of $P_{ceh-28}::gfp$ marker misexpression as one would expect in *ctbp-1* mutant AIAs. This lack of expression results in the superficially wild-type phenotype, thus appearing to suppress *ctbp-1*-dependent AIA gene repression. We speculate that loss of *ttx-3* function results in the formation of a defective AIA that is still capable of expressing some genes (such as *gcy-28.d*) and that this improperly established AIA does not attempt to express genes such as *ceh-28* that normally would be suppressed through the activity of maintenance factors like CTBP-1 (*Figure 10B*).

Perhaps unsurprisingly, we found that a cell that fails to properly establish its terminal identity does not appear to have the same needs for maintenance as does its wild-type counterpart. Further investigation of the dynamics between establishment and maintenance, and particularly characterization of mutants that help us to dissect the fine boundary between the two, will likely prove invaluable to expanding our understanding of development.

## CTBP-1 likely utilizes additional transcription factors besides EGL-13 to maintain the AIA cell identity

Our understanding of how CTBP-1 acts to maintain the AIA cell identity is incomplete. While we have identified a few genes with expression that changes in the absence of *ctbp-1* (e.g. *ceh-28*, *acbp-6*, *sra-11*, *glr-2*), none of these genes seems to individually account for the full range of AIA defects seen in older *ctbp-1* mutants. We speculate that there are many more unidentified transcriptional changes occurring in *ctbp-1* mutant AIAs that contribute to the observed AIA morphological and functional defects.

Our observations suggest that EGL-13 is not the sole transcription factor through which CTBP-1 functions to maintain AIA cell identity – neither AIA morphological defects nor some AIA gene-expression defects (i.e. *sra-11* and *glr-2* expression) in *ctbp-1* mutants were suppressed in *egl-13 ctbp-1* double mutants (*Figure 6E–H*; *Figure 8B–D*), and disruption of the EGL-13 – CTBP-1 interaction was alone not sufficient to induce AIA gene expression defects (*Figure 7C*). We propose that CTBP-1 maintains different aspects of AIA cell identity through interactions with multiple different transcription factors. Given CTBP-1's known function as a transcriptional corepressor and EGL-13's observed role in driving gene misexpression in the absence of *ctbp-1*, we speculate that CTBP-1 likely utilizes not just EGL-13 but also other transcription factors (possibly through interacting with PXDLS-like motifs located in those transcription factors) to target multiple specific DNA sequences for transcriptional silencing, effectively turning these transcription factors into transcriptional repressors. Furthermore, the decrease in *egl-13* expression in AIAs over the course of larval development suggests that CTBP-1 might regulate different transcription factors at different stages during the maintenance of the AIA cell identity. When *ctbp-1* is absent, these unregulated transcription factors can aberrantly function as transcriptional activators, resulting in either the direct or indirect expression of genes which can in turn lead to defects in other aspects of cell identity. Such a mechanism for the selective and continuous silencing of multiple genetic loci in cell-type-specific contexts by a transcriptional corepressor like CTBP-1 might explain how the broad activating activities of terminal selectors are restricted in the context of maintaining the identities of distinct cell types.

## CTBP-1 likely maintains the identities of other cells besides that of the AIAs

Others have previously reported a near pan-neuronal expression pattern of *ctbp-1* in *C. elegans* (*Reid et al., 2014*), suggesting that *ctbp-1* might be acting in more cells than just the AIAs to maintain cell identities. Why then have we thus far only been able to identify defects in the maintenance of the AIA identity in *ctbp-1* mutants? We speculate that, like the relatively subtle defects we have observed in AIA gene expression, morphology and function, *ctbp-1* mutant defects in the maintenance of other cell identities might be similarly subtle and easily missed if not specifically sought. In addition, the AIAs might be particularly susceptible to perturbations of maintenance of their identity, with defects manifesting either earlier in the life of the cell or in more distinct ways (e.g. more gene misexpression).

Both our findings and the work of others (*Reid et al., 2015*; *Sherry et al., 2020*) provide further support for the hypothesis that *ctbp-1* maintains other cell identities besides that of the AIAs. We observed that *ctbp-1* mutants have an additional AIA-independent chemotaxis defect (*Figure 4B–E*; *Figure 4—figure supplement 1A-B*), suggesting that other cells, likely neurons that sense and/or execute responses to volatile odors, are also dysfunctional. Additionally, others have shown that in *ctbp-1* mutants another pair of *C. elegans* neurons, the SMDDs, display late-onset morphological abnormalities coupled with a defect in *C. elegans* foraging behavior associated with these cells (*Reid et al., 2015*; *Sherry et al., 2020*), indicating that CTBP-1 might act to maintain SMDD cell identity as well. The broad expression of *ctbp-1* throughout much of the *C. elegans* nervous system is also consistent with the hypothesis that *ctbp-1* functions broadly to maintain multiple neuronal cell identities (*Reid et al., 2014*).

## Transcriptional corepressors might function broadly in the maintenance of cell identities

The neuron-specific expression of *ctbp-1* (*Reid et al., 2014*) suggests that CTBP-1 likely does not function in maintaining the identities of non-neuronal cells. How might non-neuronal cell identities be maintained? We speculate that transcriptional corepressors function in maintaining cell identities in both neuronal and non-neuronal cells. There are known tissue-specific activities of other corepressor complexes, such as those of NCoR1 in mediating the downstream effects of hormone sensation in the mammalian liver (*Feng et al., 2001*; *Mottis et al., 2013*) or of Transducin-Like Enhancer of Split (TLE) proteins in regulating gene expression and chromatin state in the developing mouse heart and kidney (*Sharma et al., 2009*; *Kaltenbrun et al., 2013*; *Agarwal et al., 2015*). We propose that, by analogy to CTBP-1, distinct transcriptional corepressors might specialize in the maintenance of a wide range of cell identities in distinct tissue types throughout metazoa.

# Materials and methods

## *C. elegans* strains and transgenes

All *C. elegans* strains were grown on Nematode Growth Medium (NGM) plates seeded with *E. coli* OP50 as described previously (*Brenner, 1974*). We used the N2 Bristol strain as wild type. Worms were grown at 20 °C unless otherwise indicated. Standard molecular biology and microinjection methods, as previously described (*Mello et al., 1991*), were used to generate transgenic worms.

## Plasmid construction

The *nIs175[P_{ceh-28}::gfp]*, *nIs177[P_{ceh-28}::gfp]* and *nIs348[P_{ceh-28}::mCherry]* transgenes have been previously described (*Hirose et al., 2010*). *nIs743[P_{gcy-28.d}::ctbp-1(+)]* contains 3.0 kb of the 5′ promoter of *gcy-28.d* fused to the *ctbp-1a* coding region inserted into plasmid pPD49.26. *nIs840[P_{gcy-28.d}::gfp]* contains 3.0 kb of the 5′ promoter of *gcy-28.d* inserted into pPD95.77. *nIs843[P_{gcy-28.d}::mCherry]* contains 3.0 kb of the 5′ promoter of *gcy-28.d* inserted into pPD122.56 containing mCherry. *nEx2351[P_{hsp-16.2}::ctbp-1(+); P_{hsp-16.41}::ctbp-1(+)]* contains *ctbp-1a* cDNA, isolated by RT-PCR, inserted into pPD49.78 and pPD49.83. *nEx3055[P_{gcy-28.d}::egl-13(+)]* contains 3.0 kb of the 5′ promoter of *gcy-28.d* fused to the *egl-13* coding region inserted into pPD49.26. *nEx3081[P_{acbp-6}::gfp]* contains 2.0 kb of the 5′ promoter of *acbp-6* inserted into pPD122.56. *nEx3083[P_{egl-13}::gfp]* contains 3.0 kb of the 5′ promoter of *egl-13* inserted into pPD122.56. *nIs753[P_{gcy-28.d}::ceh-28(+)]* contains 3.0 kb of the 5′ promoter of *gcy-28.d* fused to the *ceh-28* cDNA inserted into plasmid pPD49.26. AD::*ctbp-1* (used in the Y2H

assay) contains the *ctbp-1a* cDNA fused 3' of the GAL4 activation domain in the plasmid pGADT7. BD::*egl-13* and BD::*egl-13(PLNHS)* contain either wild-type *egl-13* cDNA (BD::*egl-13*) or *egl-13a* cDNA with residue 259 mutated to histidine (BD::*egl-13(PLNHS)*) fused 3' of the GAL4 DNA binding domain in the plasmid pGBKT7. Plasmid construction was performed using Infusion cloning enzymes (Takara Bio, Mountain View, CA).

## Mutagenesis screens

*ctbp-1* mutants were isolated from genetic screens for mutations that cause the survival of the M4 sister cell as scored by extra GFP-positive cells carrying the M4-cell-specific markers *nIs175[P$_{ceh-28}$::gfp]* or *nIs177[P$_{ceh-28}$::gfp]* (*Hirose and Horvitz, 2013*; *Hirose et al., 2010*). *egl-13* and *ttx-3* mutants were isolated from genetic screens for mutations that suppress *nIs175* misexpression in the AIAs of *ctbp-1(n4784)* mutants while retaining GFP expression in the M4 neuron. For both screens, mutagenesis was performed with ethyl methanesulfonate (EMS) as previously described (*Brenner, 1974*). Mutagenized P0 animals were allowed to propagate, and their F2 progeny were synchronized by hypochlorite treatment and screened at the L4 stage for extra GFP-positive cells (*ctbp-1* screens) or fewer GFP-positive cells (suppressor screens) on a dissecting microscope equipped to examine fluorescence. From both screens, mutant alleles were grouped into functional groups by complementation testing when possible. Mutants were mapped using SNP mapping (*Davis et al., 2005*) by crossing mutants to strains containing *nIs175*, *nIs177*, or *nIs175; ctbp-1(n4784)* introgressed into the Hawaiian strain CB4856. Whole-genome sequencing was performed on mutants and a combination of functional groupings and mapping data suggested genes with mutations that were likely causal for the mutant phenotypes. Rescue of mutant phenotypes with wild-type *ctbp-1(+)*, *egl-13(+)*, and *ttx-3(+)* constructs as well as the mutant phenotype of a separately isolated allele of *ctbp-1*, *tm5512*, or *ttx-3*, *ks5* and *ot22*, confirmed the identities of the causal mutations.

## Microscopy

All images were obtained using an LSM 800 confocal microscope (Zeiss LSM 800 with Airyscan Microscope, RRID:SCR_015963) and ZEN software. Images were processed and prepared for publication using FIJI software (Fiji, RRID:SCR_002285) and Adobe Illustrator (Adobe Illustrator, RRID:SCR_010279).

## Heat-shock assays

Rescue of AIA defects in day 1 adult worms was assayed using the *nEx2351[P$_{hsp-16.2}$::ctbp-1; P$_{hsp-16.41}$::ctbp-1]* transgene. Worms were synchronized and grown at 20 °C. Subsets of L1 and L4 worms carrying *nEx2351* were removed from this population for scoring at the appropriate stages. At the L4 stage, half of the worms were heat-shocked at 34 °C for 30 min and returned to 20 °C for 24 hrs while the other half remained at 20 °C throughout. After 24 hrs, heat-shocked and non-heat-shocked worms carrying *nEx2351* were scored.

## Single-cell RNA-sequencing

### Dissociation of animals into cell suspensions

Single-cell suspensions were generated as described (*Kaletsky et al., 2016*; *Taylor et al., 2019*; *Zhang and Kuhn, 2013*) with minor modifications. Briefly, synchronized populations of worms were grown on NGM plates seeded with OP50 to the L4 larval stage. Worms were harvested from these plates, washed three times with M9 buffer and treated with SDS-DTT (200 mM DTT, 0.25% SDS, 20 mM HEPES, 3% sucrose, pH 8.0) for 2–3 min. Worms were washed five times with 1 x PBS and treated with pronase (15 mg/mL) for 20–23 min. During the pronase treatment, worm suspensions were pipetted with a P200 pipette rapidly for four sets of 80 repetitions. The pronase treatment was stopped by the addition of L-15–10 media (90% L-15 media, 10% FBS). The suspension was then passed through a 35 µm nylon filter into a collection tube, washed once with 1 x PBS, and prepared for FACS.

### FACS of fluorescently labeled neurons

FACS was performed using a BD FACSAria III cell sorter running BD FACS Diva software (BD FACSARIA III cell sorter, RRID:SCR_016695). DAPI was added to samples at a final concentration of 1 µg/mL to label dead and dying cells. GFP-positive, DAPI-negative neurons were sorted from the single-cell

suspension into 1 x PBS containing 1% FBS. Non-fluorescent and single-color controls were used to set gating parameters. Cells were then concentrated and processed for single-cell sequencing.

## Single-cell sequencing
Samples were processed for single-cell sequencing using the 10X Genomics Chromium 3'mRNA-sequencing platform. Libraries were prepared using the Chromium Next GEM Single Cell 3' Kit v3.1 according to the manufacturer's protocol. The libraries were sequenced using an Illumina NextSeq 500 with 75 bp paired end reads.

## Single-cell RNA-sequencing data processing
Data processing was performed using 10X Genomics' CellRanger software (v4.0.0) (Cell Ranger, RRID:SCR_017344). Reads were mapped to the *C. elegans* reference genome from Wormbase, version WBcel235. For visualization and analysis of data, we used 10X Genomics' Loupe Browser (v4.2.0) (Loupe Browser, RRID:SCR_018555). AIAs were identified by expression of multiple AIA markers confirmed to be expressed in both wild-type and *ctbp-1* mutant AIAs (i.e. *gcy-28, ins-1, cho-1*; *Figure 2B*). Candidate genes for misexpression (either ectopic or missing) in mutant AIAs were identified and tested as described in the text.

## Morphology scoring
We assayed AIA morphology by visualizing and imaging AIAs expressing *nIs840* using an LSM 800 confocal microscope (Zeiss LSM 800 with Airyscan Microscope, RRID:SCR_015963) and a 63 x objective. AIA cell body length and area were quantified using FIJI software (Fiji, RRID:SCR_002285).

## Image blinding and scoring
A subset of 60 wild-type and 60 *ctbp-1* mutant images per stage (randomly chosen from the existing images taken to measure AIA cell body length) were selected and the genotype of each was blinded. Blinded images were then scored as either 'Normal' or 'Elongated' in appearance in batches of 40 images (20 each of wild-type and *ctbp-1* mutant, randomly assorted), repeated three times per stage. Scored images were then matched back to their genotypes and percentage of AIAs scored as 'Elongated' per genotype was calculated and graphed.

## Behavioral assays

### Butanone adaptation
Assay conditions were adapted from *Cho et al., 2016*. Staged worms were washed off non-crowded NGM plates seeded with *E. coli* OP50 with S basal. Worms were washed two times with S basal and split evenly into the 'naïve' and 'conditioned' populations. Naïve worms were incubated in 1 mL S basal for 90 min. Conditioned worms were incubated in 1 mL S basal with 2-butanone diluted to a final concentration of 120 µM for 90 min. During conditioning, unseeded NGM plates were spotted with two 1 µL drops of 10% ethanol ('control') and two 1 µL drops of 2-butanone diluted in 10% ethanol at 1:1000 ('odor') as well as four 1 µL drops of 1 M NaN$_3$ at the same loci. After conditioning, both populations were washed three more times in S basal and placed at the center of the unseeded NGM plates. Worms were allowed to chemotax for 2 hrs. Plates were moved to 4 °C for 30–60 min to stop the assay and then scored. Worms that had left the origin were scored as chemotaxing to the odor spots ('#odor') or control spots ('#control'), and a chemotaxis index was determined as (#odor - #control) / (#odor + #control). Assays were repeated on at least three separate days with one to three plates per strain ran in parallel on any given day based on the number of appropriately-staged worms available. Plates in which fewer than 50 worms left the origin were not scored.

### Chemotaxis assays
L4 worms were washed off non-crowded NGM plates seeded with *E. coli* OP50 with S basal. Worms were washed three times with S basal. Unseeded NGM plates were spotted with two 1 µL drops of 100% ethanol ('control') and two 1 µL drops of diacetyl diluted in 100% ethanol at 1:1000 or two 1 µL drops of isoamyl alcohol diluted in 100% ethanol at 1:100 ('odor') as well as four 1 µL drops of 1 M NaN$_3$ at the same loci. Worms were placed at the center of the unseeded NGM plates. Worms were

allowed to chemotax for two hrs. Plates were moved to 4 °C for 30–60 min to stop the assay and then scored. Worms that had left the origin were scored, and a chemotaxis index was determined as above. Assays were repeated on at least three separate days. Plates in which fewer than 40 worms left the origin were not scored.

## Yeast 2-hybrid assays

Fresh ( < 1 week old) Y2HGold (Takara) *S. cerevisiae* competent cells were cultured in YPDA media at 30 °C to an $OD_{600}$ between 0.4 and 0.6, harvested, washed, and resuspended in SORB buffer (110 mM TE buffer, 110 mM LiAc, 1 M Sorbital). 100 ng each of bait (pGBKT7) and prey (pGADT7) plasmids were mixed with 50 µg denatured salmon sperm carrier DNA and transformed into Y2HGold competent cells. Cells were plated on SD -Trp -Leu dropout plates and grown at 30 °C for 2–3 days until colonies were sufficiently large. Colonies from these plates were cultured in SD -Trp -Leu media at 30 °C overnight. These cultures were diluted to an $OD_{600}$ of 0.1, grown until at an $OD_{600}$ of approximately 0.5, then harvested and resuspended in 0.9% NaCl. This cell suspension was serial diluted at 1:3 with 0.9% NaCl and 3 µL of each dilution was spotted on SD -Trp -Leu and SD -His -Ade -Trp -Leu + 10 mM 3-AT dropout plates. The plates were incubated at 30 °C for 2 days and then imaged.

## CRISPR

CRISPR mutation of *egl-13* was performed according to published protocols (**Dickinson and Goldstein, 2016**). Briefly, Cas9 protein, *egl-13* guide RNA, and *egl-13* repair template were injected into MT15670 (*nIs175[P$_{ceh-28}$::gfp]*) worms alongside *dpy-10* guide RNAs (used as a coCRISPR marker). Dumpy animals were selected from the F1 generation, sequenced to confirm presences of the desired mutation in *egl-13*, then backcrossed to MT15670 worms to remove background and *dpy-10* mutations. Backcrossed strains were again genotyped to confirm the presence of the *egl-13* mutation.

 *egl-13* guide RNA sequence: GTGTCTTTTGAAAGATTTAA.

 *egl-13* repair template sequence: GGAGATTGTGGAATAGCAGTTGGAGATGGGGTGTCTTTTG AATGATTTAAAGGTGTCTCCACTTTTTCGACTGTTTGCATGTTTCCAGCGGCTGCAAGTT.

## Statistical analyses

Unpaired t-tests were used for the comparisons of AIA gene expression and AIA morphological features. One-way ANOVA tests with Tukey's correction were used for comparisons of AIA gene expression, morphological features and for behavioral assays. Statistical tests were performed using GraphPad Prism software (GraphPad Prism version 6.0 h, RRID:SCR_002798).

## Acknowledgements

We thank N An, R Droste, S Mitani, and the *Caenorhabditis* Genetics Center (CGC), which is funded by NIH Office of Research Infrastructure Programs (P40 OD010440), for strains and reagents. We thank C Diehl, E Lee, C Pender, S Sando, V Dwivedi, K Burkhart, S Wong, C Fincher, C Cho, G Johnson, D Lee, D Ghosh, A Amon, P Reddien, and Horvitz laboratory members for discussion and advice.

# Additional information

### Competing interests

The authors declare that no competing interests exist.

### Funding

| Funder | Grant reference number | Author |
| --- | --- | --- |
| Howard Hughes Medical Institute | | Josh Saul<br>Takashi Hirose<br>H Robert Horvitz |

| Funder | Grant reference number | Author |
| --- | --- | --- |
| National Institutes of Health | GM024663 | Josh Saul<br>Takashi Hirose<br>H Robert Horvitz |
| National Institutes of Health | T32GM007287 | Josh Saul |
| Friends of the McGovern Institute Fellowship | 2733360 | Josh Saul |

The funders had no role in study design, data collection and interpretation, or the decision to submit the work for publication.

### Author contributions
Josh Saul, Conceptualization, Data curation, Formal analysis, Investigation, Methodology, Validation, Visualization, Writing - original draft, Writing – review and editing; Takashi Hirose, Conceptualization, Data curation, Formal analysis, Investigation, Project administration, Validation, Visualization, Writing – review and editing; H Robert Horvitz, Conceptualization, Funding acquisition, Project administration, Supervision, Writing – review and editing

### Author ORCIDs
Josh Saul http://orcid.org/0000-0003-4193-497X
H Robert Horvitz http://orcid.org/0000-0002-9964-9613

### Decision letter and Author response
Decision letter https://doi.org/10.7554/eLife.74557.sa1
Author response https://doi.org/10.7554/eLife.74557.sa2

## Additional files

### Supplementary files
• Transparent reporting form

### Data availability
Sequencing data have been deposited in GEO under accession code GSE179484. All source data, images, numerical data and graphics have been uploaded to figshare.com and are available at https://doi.org/10.6084/m9.figshare.c.5771480.v1. All source numerical data and graphics for all figures and figure supplements have been provided.

The following dataset was generated:

| Author(s) | Year | Dataset title | Dataset URL | Database and Identifier |
| --- | --- | --- | --- | --- |
| Saul J, Horvitz HR, Hirose T | 2022 | The transcriptional corepressor CTBP-1 acts with the SOX family transcription factor EGL-13 to maintain AIA interneuron cell identity in C. elegans | https://www.ncbi.nlm.nih.gov/geo/query/acc.cgi?acc=GSE179484 | NCBI Gene Expression Omnibus, GSE179484 |

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

# Appendix 1

All strains generated in this study are available upon request from the Horvitz Laboratory.

**Appendix 1—key resources table**

| Reagent type (species) or resource | Designation | Source or reference | Identifiers | Additional information |
|---|---|---|---|---|
| Strain, strain background (*C. elegans*) | N2 (wild type) | Horvitz Lab collection | WBStrain00000001 | wild type |
| Strain, strain background (*C. elegans*) | MT15670 | Takashi Hirose/Bob Horvitz | n/a | *nIs175[P<sub>ceh-28</sub>::gfp]* |
| Strain, strain background (*C. elegans*) | MT15672 | Takashi Hirose/Bob Horvitz | n/a | *nIs177[P<sub>ceh-28</sub>::gfp]* |
| Strain, strain background (*C. elegans*) | MT15677 | This paper | n/a | *nIs175; ctbp-1(n4778)* **Figure 1—figure supplement 1A–B** |
| Strain, strain background (*C. elegans*) | MT16225 | This paper | n/a | *nIs175; ctbp-1(n4784)* **Figure 1—figure supplement 1A–B** |
| Strain, strain background (*C. elegans*) | MT15688 | This paper | n/a | *nIs175; ctbp-1(n4789)* **Figure 1—figure supplement 1A–B** |
| Strain, strain background (*C. elegans*) | MT15801 | This paper | n/a | *nIs175; ctbp-1(n4800)* **Figure 1—figure supplement 1A–B** |
| Strain, strain background (*C. elegans*) | MT15805 | This paper | n/a | *nIs175; ctbp-1(n4804)* **Figure 1—figure supplement 1A–B** |
| Strain, strain background (*C. elegans*) | MT15806 | This paper | n/a | *nIs175; ctbp-1(n4805)* **Figure 1—figure supplement 1A–B** |
| Strain, strain background (*C. elegans*) | MT15809 | This paper | n/a | *nIs175; ctbp-1(n4808)* **Figure 1—figure supplement 1A–B** |
| Strain, strain background (*C. elegans*) | MT15811 | This paper | n/a | *nIs175; ctbp-1(n4810)* **Figure 1—figure supplement 1A–B** |
| Strain, strain background (*C. elegans*) | MT15813 | This paper | n/a | *nIs175; ctbp-1(n4813)* **Figure 1—figure supplement 1A–B** |
| Strain, strain background (*C. elegans*) | MT15820 | This paper | n/a | *nIs175; ctbp-1(n4819)* **Figure 1—figure supplement 1A–B** |
| Strain, strain background (*C. elegans*) | MT15824 | This paper | n/a | *nIs175; ctbp-1(n4823)* **Figure 1—figure supplement 1A–B** |
| Strain, strain background (*C. elegans*) | MT15825 | This paper | n/a | *nIs175; ctbp-1(n4824)* **Figure 1—figure supplement 1A–B** |
| Strain, strain background (*C. elegans*) | MT15841 | This paper | n/a | *nIs177; ctbp-1(n4840)* **Figure 1—figure supplement 1A–B** |
| Strain, strain background (*C. elegans*) | MT15850 | This paper | n/a | *nIs177; ctbp-1(n4849)* **Figure 1—figure supplement 1A–B** |

*Appendix 1 Continued on next page*

*Appendix 1 Continued*

| Reagent type (species) or resource | Designation | Source or reference | Identifiers | Additional information |
|---|---|---|---|---|
| Strain, strain background (*C. elegans*) | MT15853 | This paper | n/a | *nIs177; ctbp-1(n4852)*<br>**Figure 1—figure supplement 1A–B** |
| Strain, strain background (*C. elegans*) | MT15862 | This paper | n/a | *nIs177; ctbp-1(n4861)*<br>**Figure 1—figure supplement 1A–B** |
| Strain, strain background (*C. elegans*) | MT15865 | This paper | n/a | *nIs177; ctbp-1(n4864)*<br>**Figure 1—figure supplement 1A–B** |
| Strain, strain background (*C. elegans*) | MT15866 | This paper | n/a | *nIs177; ctbp-1(n4865)*<br>**Figure 1—figure supplement 1A–B** |
| Strain, strain background (*C. elegans*) | MT26446 | This paper | n/a | *nIs175; ctbp-1(tm5512)*<br>**Figure 1—figure supplement 1C–D** |
| Strain, strain background (*C. elegans*) | MT15918 | This paper | n/a | *nIs175* introgressed into CB4856 "Hawaiian" background<br>Used to map mutants<br>**Figure 1—figure supplement 1A** |
| Strain, strain background (*C. elegans*) | MT16295 | This paper | n/a | *nIs177* introgressed into CB4856 background<br>Used to map mutants<br>**Figure 1—figure supplement 1A** |
| Strain, strain background (*C. elegans*) | MT26522 | This paper | n/a | *nIs175; ctbp-1(n4784)* introgressed into CB4856 background<br>Used to map mutants<br>**Figure 6A–C** |
| Strain, strain background (*C. elegans*) | MT23360 | This paper | n/a | *nIs175; ctbp-1(n4784); nEx2346[ctbp-1(+)]*<br>**Figure 1—figure supplement 2A** |
| Strain, strain background (*C. elegans*) | MT23361 | This paper | n/a | *nIs175; ctbp-1(n4784); nEx2347[ctbp-1(+)]*<br>**Figure 1A** |
| Strain, strain background (*C. elegans*) | MT23714 | This paper | n/a | *nIs175; ctbp-1(n4784); nIs743[P$_{gcy-28.d}$::ctbp-1(+)]*<br>**Figure 1F** |
| Strain, strain background (*C. elegans*) | MT25271 | This paper | n/a | *nIs843[P$_{gcy-28.d}$::mCherry]*<br>**Figure 1B** |
| Strain, strain background (*C. elegans*) | MT26437 | This paper | n/a | *nIs175; ctbp-1(n4784); nIs843*<br>**Figure 1B** |
| Strain, strain background (*C. elegans*) | MT23365 | This paper | n/a | *nIs175; ctbp-1(n4784); nEx2351[P$_{hsp-16.2}$::ctbp-1(+);P$_{hsp-16.41}$::ctbp-1(+)]*<br>**Figure 1H** |
| Strain, strain background (*C. elegans*) | MT18778 | Takashi Hirose/Bob Horvitz | n/a | *nIs348[P$_{ceh-28}$::mCherry]; lin-15AB(n765ts)* |
| Strain, strain background (*C. elegans*) | MT20844 | This paper | n/a | *nIs348[P$_{ceh-28}$::mCherry]; ctbp-1(n4784)*<br>**Figure 1—figure supplement 1E** |
| Strain, strain background (*C. elegans*) | NH2466 | *Caenorhabditis* Genetics Center (CGC) | WBStrain00028771 | *ayIs4[P$_{egl-17}$::gfp]; dpy-20(e1282ts)* |
| Strain, strain background (*C. elegans*) | MT26417 | This paper | n/a | *ayIs4; nIs348; ctbp-1(n4784)*<br>**Figure 2A** |

*Appendix 1 Continued on next page*

*Appendix 1 Continued*

| Reagent type (species) or resource | Designation | Source or reference | Identifiers | Additional information |
|---|---|---|---|---|
| Strain, strain background (*C. elegans*) | BW1946 | *Caenorhabditis* Genetics Center (CGC) | WBStrain00004003 | *ctIs43[P$_{dbl-1}$::gfp] unc-42(e270)* |
| Strain, strain background (*C. elegans*) | MT23726 | This paper | n/a | *nIs348; ctIs43 unc-42(e270); ctbp-1(n4784)* **Figure 2A** |
| Strain, strain background (*C. elegans*) | MT20852 | This paper | n/a | *nIs491[P$_{ser-7.b}$::mCherry]* **Figure 2A** |
| Strain, strain background (*C. elegans*) | MT23427 | This paper | n/a | *nIs491; ctbp-1(n4784)* **Figure 2A** |
| Strain, strain background (*C. elegans*) | NY2080 | *Caenorhabditis* Genetics Center (CGC) | WBStrain00029170 | *ynIs80[P$_{flp-21}$::gfp]* |
| Strain, strain background (*C. elegans*) | MT23718 | This paper | n/a | *nIs348; ctbp-1(n4784); ynIs80* **Figure 2A** |
| Strain, strain background (*C. elegans*) | OH10237 | *Caenorhabditis* Genetics Center (CGC) | WBStrain00029598 | *otIs326[P$_{ins-1}$::gfp]* |
| Strain, strain background (*C. elegans*) | MT26422 | This paper | n/a | *ctbp-1(n4784); otIs326* **Figure 2B** |
| Strain, strain background (*C. elegans*) | JN1716 | *Caenorhabditis* Genetics Center (CGC) | n/a | *peIs1716[P$_{ins-1s}$::gfp;P$_{ttx-3}$::mCherry]* |
| Strain, strain background (*C. elegans*) | MT23717 | This paper | n/a | *nIs348; ctbp-1(n4784); peIs1716* **Figure 2B** |
| Strain, strain background (*C. elegans*) | OH11030 | *Caenorhabditis* Genetics Center (CGC) | WBStrain00029645 | *otIs317[P$_{mgl-1}$::mCherry]; otIs379[P$_{cho-1}$::gfp]* |
| Strain, strain background (*C. elegans*) | MT26421 | This paper | n/a | *nIs348; ctbp-1(n4784); otIs317; otIs379* **Figure 2B** |
| Strain, strain background (*C. elegans*) | MT26420 | This paper | n/a | *ctbp-1(n4784); otIs317* **Figure 2B** |
| Strain, strain background (*C. elegans*) | MT25268 | This paper | n/a | *nIs840[P$_{gcy-28.d}$::gfp]* **Figure 3A–D** |
| Strain, strain background (*C. elegans*) | MT25270 | This paper | n/a | *nIs842[P$_{gcy-28.d}$::gfp]* **Figure 3A–D** |
| Strain, strain background (*C. elegans*) | MT26412 | This paper | n/a | *nIs348; ctbp-1(n4784); nIs840* **Figure 3A–D** |
| Strain, strain background (*C. elegans*) | MT26438 | This paper | n/a | *nIs348; ctbp-1(n4784); nIs743; nIs840* **Figure 3E–H** |
| Strain, strain background (*C. elegans*) | MT26439 | This paper | n/a | *nIs348; ctbp-1(n4784); nIs840; nEx2351* **Figure 3I–L** |
| Strain, strain background (*C. elegans*) | JN580 | *Caenorhabditis* Genetics Center (CGC) | n/a | *peIs580[P$_{ins-1s}$::casp1;P$_{ins-1s}$::venus;P$_{unc-122}$::gfp]* |

*Appendix 1 Continued on next page*

*Appendix 1 Continued*

| Reagent type (species) or resource | Designation | Source or reference | Identifiers | Additional information |
|---|---|---|---|---|
| Strain, strain background (*C. elegans*) | MT23746 | This paper | n/a | *nIs175; egl-13(n5937) ctbp-1(n4784)* **Figure 6B** |
| Strain, strain background (*C. elegans*) | MT24129 | This paper | n/a | *nIs175; egl-13(n6013) ctbp-1(n4784)* **Figure 6B** |
| Strain, strain background (*C. elegans*) | MT25352 | This paper | n/a | *nIs175; egl-13(n6313) ctbp-1(n4784)* **Figure 6B** |
| Strain, strain background (*C. elegans*) | MT25347 | This paper | n/a | *nIs175; ctbp-1(n4784) ttx-3(n6308)* **Figure 6C** |
| Strain, strain background (*C. elegans*) | MT25355 | This paper | n/a | *nIs175; ctbp-1(n4784) ttx-3(n6316)* **Figure 6C** |
| Strain, strain background (*C. elegans*) | MT26486 | This paper | n/a | *nIs175; egl-13(n5937) ctbp-1(n4784); nEx3062[egl-13(+)]* **Figure 6—figure supplement 1C** |
| Strain, strain background (*C. elegans*) | MT26487 | This paper | n/a | *nIs175; egl-13(n5937) ctbp-1(n4784); nEx3063[egl-13(+)]* **Figure 6—figure supplement 1C** |
| Strain, strain background (*C. elegans*) | MT26549 | This paper | n/a | *nIs175; egl-13(n6013) ctbp-1(n4784); nEx3080[egl-13(+)]* **Figure 6—figure supplement 1C** |
| Strain, strain background (*C. elegans*) | MT26523 | This paper | n/a | *nIs175; egl-13(n6313) ctbp-1(n4784); nEx3074[egl-13(+)]* **Figure 6—figure supplement 1C** |
| Strain, strain background (*C. elegans*) | MT26548 | This paper | n/a | *nIs175; egl-13(n6313) ctbp-1(n4784); nEx3079[egl-13(+)]* **Figure 6—figure supplement 1C** |
| Strain, strain background (*C. elegans*) | MT26448 | This paper | n/a | *nIs175; ctbp-1(n4784) ttx-3(ot22)* **Figure 6—figure supplement 2B** |
| Strain, strain background (*C. elegans*) | MT26447 | This paper | n/a | *nIs175; ctbp-1(n4784) ttx-3(ks5)* **Figure 6—figure supplement 2B** |
| Strain, strain background (*C. elegans*) | MT26491 | This paper | n/a | *nIs175; ctbp-1(n4784) ttx-3(n6308); nEx3067[ttx-3(+)]* **Figure 6—figure supplement 2C** |
| Strain, strain background (*C. elegans*) | MT26492 | This paper | n/a | *nIs175; ctbp-1(n4784) ttx-3(n6308); nEx3068[ttx-3(+)]* **Figure 6—figure supplement 2C** |
| Strain, strain background (*C. elegans*) | MT26493 | This paper | n/a | *nIs175; ctbp-1(n4784) ttx-3(n6308); nEx3069[ttx-3(+)]* **Figure 6—figure supplement 2C** |
| Strain, strain background (*C. elegans*) | MT26521 | This paper | n/a | *nIs175; ctbp-1(n4784) ttx-3(n6316); nEx3073[ttx-3(+)]* **Figure 6—figure supplement 2C** |
| Strain, strain background (*C. elegans*) | MT26528 | This paper | n/a | *nIs175; ctbp-1(n4784) ttx-3(n6316); nEx3078[ttx-3(+)]* **Figure 6—figure supplement 2C** |
| Strain, strain background (*C. elegans*) | MT26481 | This paper | n/a | *nIs175; egl-13(n5937) ctbp-1(n4784); nEx3055[P$_{gcy-28.d}$::egl-13(+)]* **Figure 7—figure supplement 1A–B** |
| Strain, strain background (*C. elegans*) | MT26441 | This paper | n/a | *nIs175; egl-13(n5937) ctbp-1(n4784); nIs840* **Figure 6E** |

*Appendix 1 Continued on next page*

*Appendix 1 Continued*

| Reagent type (species) or resource | Designation | Source or reference | Identifiers | Additional information |
|---|---|---|---|---|
| Strain, strain background (*C. elegans*) | MT26442 | This paper | n/a | nIs175; ctbp-1(n4784) ttx-3(n6308); nIs840 **Figure 6E** |
| Strain, strain background (*C. elegans*) | MT26415 | This paper | n/a | evIs111[P_{rgef-1}::gfp]; nIs843 **Figure 5A–E** |
| Strain, strain background (*C. elegans*) | MT26416 | This paper | n/a | ctbp-1(n4784); evIs111; nIs843 **Figure 5A–E** |
| Strain, strain background (*C. elegans*) | MT26444 | This paper | n/a | otIs123[P_{sra-11}::gfp]; nIs843 **Figure 5C** |
| Strain, strain background (*C. elegans*) | MT26580 | This paper | n/a | nIs843; nEx3083[P_{egl-13}::gfp] **Figure 5A** |
| Strain, strain background (*C. elegans*) | MT26604 | This paper | n/a | ctbp-1(n4784); nIs843; nEx3083 **Figure 5A** |
| Strain, strain background (*C. elegans*) | MT26808 | This paper | n/a | nIs175; egl-13(n6675) **Figure 7C** |
| Strain, strain background (*C. elegans*) | MT26445 | This paper | n/a | nIs348; ctbp-1(n4784); otIs123 **Figure 5C** |
| Strain, strain background (*C. elegans*) | MT26524 | This paper | n/a | nIs348; egl-13(n5937) ctbp-1(n4784); otIs123 **Figure 8B** |
| Strain, strain background (*C. elegans*) | MT26504 | This paper | n/a | nIs843; ivEx138[P_{glr-2}::gfp] **Figure 5E** |
| Strain, strain background (*C. elegans*) | MT26505 | This paper | n/a | nIs348; ctbp-1(n4784); ivEx138 **Figure 5E** |
| Strain, strain background (*C. elegans*) | MT26550 | This paper | n/a | nIs348; egl-13(n5937) ctbp-1(n4784); ivEx138 **Figure 8C** |
| Strain, strain background (*C. elegans*) | MT26581 | This paper | n/a | nIs843; nEx3081[P_{acbp-6}::gfp] **Figure 5A** |
| Strain, strain background (*C. elegans*) | MT26551 | This paper | n/a | nIs348; ctbp-1(n4784); nEx3081 **Figure 5A** |
| Strain, strain background (*C. elegans*) | MT26582 | This paper | n/a | nIs348; egl-13(n5937) ctbp-1(n4784); nEx3081 **Figure 8A** |
| Strain, strain background (*C. elegans*) | MT26605 | This paper | n/a | acbp-6(tm2995); nIs175; ctbp-1(n4784) **Figure 9A–D** |
| Strain, strain background (*C. elegans*) | MT23725 | This paper | n/a | nIs175; ctbp-1(n4784) ceh-28(cu11) **Figure 9E–H** |
| Strain, strain background (*C. elegans*) | MT23736 | This paper | n/a | nIs753[P_{gcy-28.d}::ceh-28(+)] **Figure 9I–L** |

