## [Editor Report]

The paper presents an interesting addition to our understanding of cell fate maintenance, making incisive use of the power of *C. elegans* genetics.

---

## [Decision Letter]

[Editors' note: this paper was reviewed by Review Commons.]

**Decision letter after peer review:**

Thank you for submitting your article "The transcriptional corepressor CTBP-1 acts with the SOX family transcription factor EGL-13 to maintain AIA interneuron cell identity in *C. elegans*" for consideration by *eLife*. Your article has been reviewed by 3 peer reviewers at Review Commons, and the evaluation at *eLife* has been overseen by a Reviewing Editor and Piali Sengupta as the Senior Editor.

Based on your manuscript, the reviews and your responses, we invite you to submit a revised version incorporating the revisions as outlined in your response to the reviews. It is possible that the revised paper will require re-review.

When preparing your revisions, please also address the following points. Please note that some of these points were not raised by the previous reviewers but are requested by the editors.

1. The issue of egl-13 expression and possible regulation by CTBP-1 in AIA. This was also raised by the reviewers.

2. Please integrate the work in this manuscript better with previous results regarding mechanisms of AIA specification. In particular, it is important to integrate findings regarding the role of TTX-3 in AIA specification with the observations reported here. Is ttx-3 expression affected in egl-13 and/or ctbp-1 mutants? Does the ectopic marker expression observed in ctbp-1 mutants require ttx-3?

3. Please mutate the predicted binding site (PLNLS motif) of CTBP-1 in EGL-13 and assess effects on gene expression. Ideally, this would be done by gene editing.

4. The editors noted a few figures in the manuscript in which presumably representative images are provided but lack any quantification. Please include detailed quantification of all shown phenotypes.

---

## [Author Response]

When preparing your revisions, please also address the following points. Please note that some of these points were not raised by the previous reviewers but are requested by the editors.1. The issue of egl-13 expression and possible regulation by CTBP-1 in AIA. This was also raised by the reviewers.2. Please integrate the work in this manuscript better with previous results regarding mechanisms of AIA specification. In particular, it is important to integrate findings regarding the role of TTX-3 in AIA specification with the observations reported here. Is ttx-3 expression affected in egl-13 and/or ctbp-1 mutants? Does the ectopic marker expression observed in ctbp-1 mutants require ttx-3?3. Please mutate the predicted binding site (PLNLS motif) of CTBP-1 in EGL-13 and assess effects on gene expression. Ideally, this would be done by gene editing.4. The editors noted a few figures in the manuscript in which presumably representative images are provided but lack any quantification. Please include detailed quantification of all shown phenotypes.

We have made the following additions and revisions:

1. We tested the effect of heat shock on AIA morphology in wild-type and *ctbp-1* mutant worms. We found no significant difference in morphology between heat shocked and non-heat shocked cells.

2. We assayed *egl-13* expression using a *gfp* transcriptional reporter in wild-type and *ctbp-1* mutant worms. We found that *egl-13* is expressed in the AIAs of early larval stage worms, that this expression decreases over time, and that mutation of *ctbp-1* does not affect expression in the AIAs.

3. We tested the effect of AIA-specific *ceh-28* overexpression on AIA function. We found that this *ceh-28* overexpression disrupts AIA function at both the L1 and L4 larval stages.

4. We tested the ability of CTBP-1 and EGL-13 to physically interact using a yeast 2-hybrid assay. In this assay, we found that these two proteins can indeed physically interact and that mutation of EGL-13’s PLNLS binding motif disrupts this interaction.

5. We added information about the role of *ttx-3* and AIA specification in AIA cell-identity maintenance. We had actually identified *ttx-3* in the same suppressor screen in which we identified *egl-13* as a *ctbp-1* suppressor but had not included this information in our original manuscript to simplify our narrative. We have now added data about *ttx-3* to the manuscript, including the lack of an effect of *ctbp-1* mutation on *ttx-3* expression, characterization of *ttx-3*’s ability to partially suppress AIA morphological defects and its failure to suppress AIA functional defects. We have also included a section in the Discussion in which we speculate about how the failure to proper specify the AIA cell identity, as is likely happening in *ttx-3* mutants, would appear to suppress AIA cell-identity maintenance defects caused by a loss of *ctbp-1*.

6. We used CRISPR to mutate the PLNLS binding motif in EGL-13 and assayed the effect on *ceh-28* expression in AIA. We found that this mutation alone does not result in reporter misexpression, supporting the idea that CTBP-1 is acting with additional transcription factors to maintain the AIA cell identity.

7. We provide quantification of data corresponding to representative images in our figures in the Supplement and added language to both the main text and the figure legends to explicitly draw the reader’s attention to this quantification.

[Editors' note: we include below the reviews that the authors received from Review Commons, along with the authors’ responses.]

Reviewer #1:Major comments:The manuscript is very well written and results have been very clearly presented. The key conclusions drawn by the authors are convincing. However, one of the claims by the authors is not supported by the data. In lines 206-215 the authors discuss experiments where they visualized the morphology of the AIAs in ctbp-1 mutants where ctbp-1 expression is restored temporally in the L4-young adult stage using a heat-shock promoter construct. The authors conclude that "ctbp-1 can act.… in older worms to maintain aspects of AIA morphology in a manner similar to AIA gene expression." However, the data presented in Figure 3I-L show no statistically significant difference between ctbp-1 mutants and mutants with the HS-construct, either with and without heat shock. Thus, although there seems to be some effect of the heat shock, this is not significant and thus does not support the conclusion of the authors. In addition, an important control is missing. How does the heat shock affect the morphology of AIAs in wt or ctbp-1 animals, without the hs-construct?

We agree with this comment and have updated the manuscript to clarify that suggestion of the activity of CTBP-1 in preventing further disruption of AIA morphology is speculative. We will conduct the suggested control experiment and include the results in a revised version of the manuscript.

Apart from the above, all strong claims by the authors are valid. In addition, the authors suggest a mechanism, where CTBP-1 regulates the function of the EGL-13 transcription factor in AIA and that overexpression of CEH-28 in AIA contributes to the olfactory adaptation defect observed in the ctbp-1 mutant animals. These mechanistic speculations could be relatively easily strengthened by two additional experiments.One, does ctbp-1 loss of function affect egl-13 expression? The model presented in Figure 8 suggests that egl-13 expression levels are not affected, but from the data in the paper it is not even clear of egl-13 is expressed in AIA. Whether egl-13 is expressed in AIA, and if its expression levels are affected by mutation of ctbp-1 could be tested using egl-13::gfp expressing animals.

This is an excellent suggestion and experiments we had been attempting already. We will include findings from these experiments once they are complete in a revised version of the manuscript.

Two, does overexpression of ceh-28 cause an olfactory adaptation defect? This could be tested by cell specific overexpression of ceh-28 in AIA.

This is also a great suggestion. We will conduct this experiment and include the findings in a revised manuscript.

The data and the methods have been presented in such a way that they can be reproduced. I do have some doubts with regard to the statistical analysis. The authors report that statistical analysis involved unpaired ttests. But as all results involve the analysis of data from 3-5 different strains, a multiple sample analysis should be used. To correct for the number of samples, one should first use an ANOVA to test for statistical differences, followed by a post hoc analysis to identify those that are significantly different.

We agree with this criticism. We have replaced instances of multiple sample analyses with a one-way ANOVA test followed by Tukey’s multiple test correction. The current version of the manuscript reflects these changes in figures, figure legends and in the Materials and methods.

Reviewer #2:Major comments:1. The paper is well written and figures are clearly organized. The authors made suitable conclusions based on the data provided. Materials and methods are appropriately described for reproductivity.2. It would strengthen the model (Figure 8) by testing physical interaction between CTBP-1 and EGL-13 in AIA using BiFC.

We agree and are currently attempting such experiments. Meaningful results from these experiments will be included in a revised manuscript.

Reviewer #3 (Evidence, reproducibility and clarity (Required)):Major comments:The key conclusions of this manuscript are highly convincing and are supported by multiple mutant alleles and rescue experiments.There are certain claims in the manuscript that need to be clarified (detailed below).No additional experiments are essential to support the claims of the paper.Most of the data and the methods presented well – however a Table listing genes identified in the AIA-specific RNA Seq is required. The GEO accession number has been made available for the RNA Sequencing data however listed the genes identified would aid the reader. Were ctbp-1 and egl-13 shown to be expressed in the AIAs using this approach?

We have included such a table, replacing Figure S6 (which previously showed only *ceh-28* expression) with a table listing expression of all confirmed hits from the scRNA-Seq experiment. *ctbp-1* and *egl-13* were also found to be expressed in the AIA neurons in this scRNA-Seq experiment.

No evidence is presented that EGL-13 is expressed in the AIAs?

As noted above, the scRNA-Seq experiment showed *egl-13* expression in the AIAs. We also will assay *egl-13* expression in the AIAs using a GFP reporter and include the results in a revised manuscript.

Can the authors comment and include in the manuscript information regarding whether the promoters of AIA-expressed genes that are regulated by EGL-13 contain EGL-13 binding sites? Also, are the promoters of AIA-expressed genes not regulated by EGL-13 missing these sites?

We have added such information to the manuscript. Briefly, our analysis identified no promising candidates for EGL-13 binding sites in the promoter regions of either *ceh-28* or *acbp-6*, suggesting that regulation of these by EGL-13 is likely indirect. Further, no previous work has indicated that either of these genes is regulated directly by EGL-13, although in the case of *acbp-6* little is known about this gene or the ways in which it is regulated. However, the claim that EGL-13 regulates expression of *acbp-6* and *ceh-28* indirectly is speculative and is not a conclusion of this current work.

Experiments and statistical analysis are adequate.